# Distributionally Robust Weighted $k$-Nearest Neighbors

**Shixiang Zhu**
Carnegie Mellon University
shixianz@andrew.cmu.edu

**Liyan Xie**
The Chinese University of Hong Kong, Shenzhen
xieliyan@cuhk.edu.cn

**Minghe Zhang**
Georgia Institute of Technology
minghe_zhang@gatech.edu

**Rui Gao**
University of Texas at Austin
rui.gao@mccombs.utexas.edu

**Yao Xie**
Georgia Institute of Technology
yao.xie@isye.gatech.edu

## Abstract

Learning a robust classifier from a few samples remains a key challenge in machine learning. A major thrust of research has been focused on developing $k$-nearest neighbor ($k$-NN) based algorithms combined with metric learning that captures similarities between samples. When the samples are limited, robustness is especially crucial to ensure the generalization capability of the classifier. In this paper, we study a minimax distributionally robust formulation of weighted $k$-nearest neighbors, which aims to find the optimal weighted $k$-NN classifiers that hedge against feature uncertainties. We develop an algorithm, `Dr.k-NN`, that efficiently solves this functional optimization problem and features in assigning minimax optimal weights to training samples when performing classification. These weights are class-dependent, and are determined by the similarities of sample features under the least favorable scenarios. The proposed framework can be shown to be equivalent to a Lipschitz norm regularization problem. We also couple our framework with neural-network-based feature embedding. We demonstrate the competitive performance of our algorithm compared to the state-of-the-art in the few-training-sample setting with various real-data experiments.

## 1 Introduction

Machine learning has been proven successful in data-intensive applications but is often hampered when the data set is small. For example, in breast mammography diagnosis for breast cancer screening [5], the diagnosis of the type of breast cancer requires specialized analysis by pathologists in a highly time- and cost-consuming task and often leads to non-consensual results. As a result, labeled data in digital pathology are generally very scarce; so do many other applications.

In this paper, we aim to tackle the general multi-class classification problem when only very few training samples are available for each class [43]. Evidently, $k$-Nearest Neighbor ($k$-NN) algorithm [3] is a natural idea to tackle this problem and shows promising empirical performances. Notable contributions, including seminal work [19] and the follow-up non-linear version [34], go beyond the vanilla $k$-NN and propose neighborhood component analysis (NCA). NCA learns a distance metric that minimizes the expected leave-one-out classification error on the training data using a stochastic neighbor selection rule. Some recent studies in few-shot learning utilize the limited training data using a similar idea, such as matching network [41] and prototypical network [39]. They are primarily based on distance-weighted $k$-NN, which classifies an unseen sample (aka. *query*) by a weighted

36th Conference on Neural Information Processing Systems (NeurIPS 2022).

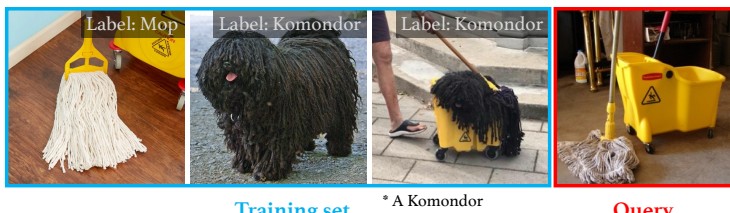

Figure 1: Motivating example: a small training set of three image-label pairs: the first image is a mop; the second image is a dog; the third image looks like a mop but is, in fact, a dog (dressing up as a mop). The query (the last image) is "closer" to the third one and more likely to be misclassified as a dog if we use distance-weighted $k$-NN.

vote of its neighbors and uses the distance between two data points in the embedding space as their weights.

The classification performance of weighted $k$-NN critically depends on the choice of weighting scheme. The distance measuring the similarity between samples is typically chosen by metric learning, where a task-specific distance metric is automatically constructed from supervised data [24, 33, 41]. However, it has been recognized that they may be not robust to the *few-training-samples* scenario, where an "outlier" may greatly deteriorate the performance. An example to illustrate this issue is shown in Figure 1. The training set with only three labeled samples includes two categories we want to classify: mop and Komondor. As we can see, the query image is visually closer to the third sample and thus more likely to be misclassified as a Komondor. Here the third sample in the training set is an "outlier", since it is a Komondor dressing up as a mop, and it misleads the metric learning model to capture irrelevant details (e.g., the bucket and the mop handle) for the Komondor category. Such a problem can become even severe when the sample size is small.

The discussion above highlights the importance of choosing a good weighting scheme in weighted $k$-NN. To develop algorithms that are more robust in the few-training-samples settings, we propose a new formulation of distributionally robust weighted $k$-nearest neighbors. More specifically, for a given set of features of training samples, we solve a Wasserstein distributionally robust optimization problem that finds the minimax optimal weight functions for the $k$-nearest neighbors. This infinite-dimensional functional optimization over weight functions presents a unique challenge for which existing literature on distributionally robust optimization do not consider. To tackle this challenge, we first consider a relaxed problem that optimizes over all randomized classifiers, which turns out to admit a finite-dimensional convex programming reformulation in spite of being infinite-dimensional (Theorem 1). Next, we show that there is a weighted $k$-NN classifier achieving the same risk and shares the same least favorable distributions (LFDs) as the robust classifier (Theorem 2). Thereby we prove the optimality of such weighted $k$-NN classifier for the original distributionally robust weighted $k$-nearest neighbors problem. Furthermore, we prove the equivalence between the proposed robust weighted $k$-NN classifier with a Lipschitz norm regularization problem (Theorem 3), which provides promising insights on the generalization property of the robust classifier.

Based on these theoretical results, we proposed a novel algorithm called `Dr.k-NN`. Unlike the traditional distance-weighted $k$-NN that uses the same weight for all label classes, our algorithm introduces a vector of weights, one for each class, for each sample in $k$-NN and performs a weighted majority vote. These weights are determined from the LFDs and reveal the significance of each sample in the worst case, thereby contributing effectively to final decision making. An example is illustrated in Figure 2. Further, using differentiable optimization [4, 2], we incorporate a neural network into the minimax classifier that jointly learns the feature embedding and the minimax optimal classifier. Numerical experiments show that our algorithm can effectively improve the multi-class classification performance with few training samples on various data sets.

**Related work** Recently, there has been much interest in multi-class classification with few training samples, see [31, 43] for a survey. The main idea of our work is related to metric learning [19, 22, 33, 34, 32], which essentially translates the hidden information carried by the limited data into a distance metric, and has been widely adopted in few-shot learning and meta learning [15, 24, 39, 41, 30]. However, unlike few-shot and meta learning, where the goal is to acquire meta knowledge from a

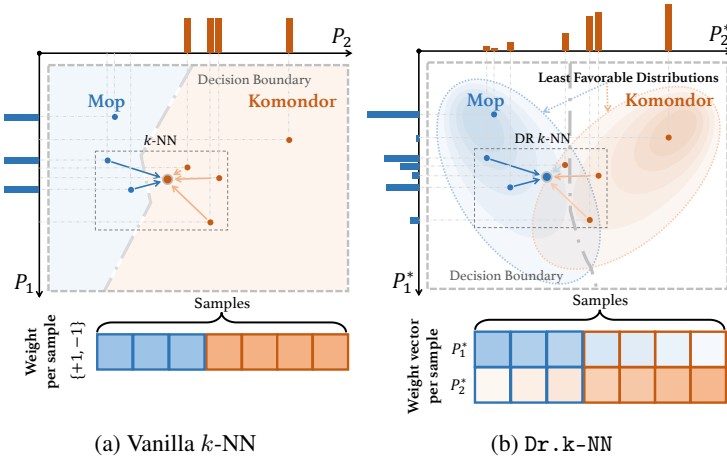

(a) Vanilla $k$-NN  (b) `Dr.k-NN`

Figure 2: An illustrative comparison of `Dr.k-NN` and vanilla $k$-NN. Each colored dot is a training sample, where the color indicates its class-membership. The horizontal/vertical bar represents the probability mass of one training sample under the distribution $P_1$, $P_2$, respectively.

large number of observed classes and then predict examples from unobserved classes, we focus on attacking a specific general classification problem where the number of categories is fixed but labeled data are scarce. In this paper, we take a different probabilistic approach to exploit information from the data: we construct an uncertainty set for distributions of each class based on the Wasserstein distance.

Wasserstein distributionally robust optimization [10, 7, 13, 1, 8, 17, 38, 6, 36, 16] is an emerging paradigm for statistical learning; see [26] for a recent survey. Our work is mostly related to [18], a framework for Wasserstein robust hypothesis testing, but is different in three important ways. First, we focus on multi-class classification, while [18] only studied two hypotheses. Second, we focus on directly minimizing the mis-classification error while [18] used a convex relaxation for the 0-1 loss. Third, we analyze the generalization bound while [18] does not. Fourth, while [18] requires sample features as an input, we develop a scalable algorithmic framework to simultaneously learn the optimal feature extractor parameterized by neural networks and robust classifier to achieve the best performance. A recent work [9] studies distributionally robust $k$-NN regression. Note that regression and classification are fundamentally different as different performance metrics are used. In [9] the objective is to minimize the mean square error, whereas in our work we minimize classification errors. Last but not least, our minimax formulation is similar to [14], which introduces a generalization of the maximum entropy principle and aims to address a different supervised learning problem. It is worth noting that the paper considers another type of ambiguity set defined via cross-moment constraints. In contrast, we adopt Wasserstein sets in our formulation, leading to different tractable optimization reformulations and regularization terms.

Another well-known work on optimal weighted nearest neighbor binary classifier [35] assigns one weight to each sample; the optimal weights minimize asymptotic expansion for the excess risk (regret). In contrast, we consider minimax robust multi-class classification, each training sample is associated with different weights for different classes, and we consider the few sample regime instead of the asymptotic regime in [35].

## 2 Distributionally Robust $k$-NN

In this section, we present our model. We first define weighted $k$-NN classifier in Section 2.1, then present the proposed framework of distributionally robust $k$-NN problem in Section 2.2.

### 2.1 Weighted $k$-NN classifier

Let $\{(x^1, y^1), \ldots, (x^n, y^n)\}$ be a set of training samples, where $x^i$ denotes the $i$-th data sample in the observation space $\mathcal{X}$, and $y^i \in \mathcal{Y} := \{1, \ldots, M\}$ denotes the class (label) of the $i$-th data sample. Let $\phi : \mathcal{X} \to \Xi$ be a feature extractor that embeds samples to the feature space $\Xi$ (in Section 4.2 we will train a neural network to learn $\phi$). Denote the sample feature vectors and the empirical support

as:
$$\xi^i := \phi(x^i), \ i = 1, \ldots, n, \quad \widehat{\Xi} := \{\xi^1, \ldots, \xi^n\}.$$

Let
$$S = \{(\xi^1, y^1), \ldots, (\xi^n, y^n)\}.$$

Define empirical distributions:
$$\widehat{P}_m := \frac{1}{|\{i : y^i = m\}|} \sum_{i=1}^n \delta_{\xi^i} \mathbb{I}\{y^i = m\}, \ m = 1, \ldots, M,$$

where $\delta$ denotes the Dirac point mass, $|\cdot|$ denotes the cardinality of a set, and $\mathbb{I}$ denotes the indicator function.

Let $\pi : \Xi \to \Delta_M$ be a *randomized* classifier that assigns class $m \in \{1, \ldots, M\}$ with probability $\pi_m(\xi)$ to a query feature vector $\xi \in \Xi$, where $\Delta_M$ is the probabilistic simplex $\Delta_M = \{\pi \in \mathbb{R}_+^M : \sum_{m=1}^M \pi_m = 1\}$. It is worth mentioning that the randomized test is more general than the commonly seen deterministic test. In particular, the random classifier $\pi$ reduces to the deterministic test if for any $\xi$, there exists a $m$ such that $\pi_m(\xi) = 1$. Suppose the features in each class $m$ follows a distribution $P_m$. We define the *risk* of a classifier $\pi$ as the total error probabilities[1]

$$\Psi(\pi; P_1, \ldots, P_M) := \sum_{m=1}^M \mathbb{E}_{\xi \sim P_m}[1 - \pi_m(\xi)]. \tag{1}$$

Recall that the vanilla $k$-NN is performed as follows. Let $c : \Xi \times \Xi \to \mathbb{R}_+$ be a metric on $\Xi$ that measures distance between features. For any given query point $\xi$, let $\tau_1^S(\xi), \ldots, \tau_n^S(\xi)$ be a reordering of $\{1, \ldots, n\}$ according to their distance to $\xi$, i.e., $c(\xi, \xi^{\tau_i^S(\xi)}) \leq c(\xi, \xi^{\tau_{i+1}^S(\xi)})$ for all $i < n$, where the tie is broken arbitrarily. Here the superscript $S$ indicates the dependence on the sample $S$. In vanilla $k$-NN, we compute the votes as

$$p_m(\xi) := \sum_{i=1}^k \frac{1}{k} \mathbb{I}\{y^{\tau_i^S(\xi)} = m\}, \ m = 1, \ldots, M. \tag{2}$$

The vanilla $k$-NN decides the class for $\xi$ by the majority vote, i.e., accept the class $\arg\max_{1 \leq m \leq M} p_m(\xi)$.

To define a weighted $k$-NN, let us replace the equal weights in (2) by an arbitrary weight function $w_m : \Xi \times \Xi \to \mathbb{R}_+$ for each class $m = 1, \ldots, M$:

$$p_m(\xi) := \sum_{i=1}^k w_m(\xi, \xi^{\tau_i^S(\xi)}), \tag{3}$$

and use a shorthand notation $w := (w_1, \ldots, w_M)$. In the sequel, we define a general tie-breaking rule as follows. For any $\xi$, denote $\mathcal{M}_0(\xi) := \arg\max_{1 \leq m \leq M} p_m(\xi)$. When $|\mathcal{M}_0(\xi)| > 1$, there is a tie at $\xi$. We denote $\pi_m(\xi)$ as the probability of accepting class $m$ for $m \in \mathcal{M}_0(\xi)$ and we have $\sum_{m \in \mathcal{M}_0(\xi)} \pi_m(\xi) = 1$.

We define a *weighted $k$-NN classifier* $\pi^{\mathsf{knn}}(\xi; k, w) : \Xi \to \Delta_M$ as:

$$\pi_m^{\mathsf{knn}}(\xi; k, w) = \begin{cases} \pi_m(\xi), & m \in \mathcal{M}_0(\xi), \\ 0, & \text{otherwise.} \end{cases} \tag{4}$$

A weighted $k$-NN classifier involves two parameters: number of nearest neighbors $k$ and weighting scheme $w$. Particularly, $w_m(\xi, \xi^i) = \mathbb{I}\{y^i = m\}$ recovers the vanilla $k$-NN, and $w_m(\xi, \xi^i) = \mathbb{I}\{y^i = m\}/c(\xi, \xi^i)$ recovers the distance-based weighted $k$-NN. Note that our definition (3) allows different weighting schemes for different classes, which is more general than the standard weighted $k$-NN.

The goal is to find the optimal weighted $k$-NN classifier $\pi^{\mathsf{knn}}(\cdot; k, w)$ such that the risk $\Psi$ as defined in (1) is minimized. Since the underlying true distributions are unknown, the commonly used loss

---

[1]To ease the exposition we consider only equal weights over the error probabilities, but our results can be easily generalized to any weighted average of error probabilities.

function is the empirical loss, i.e., substitute the empirical distributions $\widehat{P}_m$ into the risk function (1). This leads to the following optimization problem:

$$\min_{\substack{1 \leq k \leq n \\ w_m : \Xi \times \Xi \to \mathbb{R}_+, \, 1 \leq m \leq M}} \Psi(\pi^{\mathsf{knn}}(\cdot; k, w); \widehat{P}_1, \ldots, \widehat{P}_M). \tag{5}$$

It is worth mentioning that this minimization problem is an *infinite-dimensional* functional optimization, since the weighting schemes $w$ is a function on $\Xi \times \Xi$.

## 2.2 Distributionally robust $k$-NN

For few-training-sample setting, the empirical distributions might not be good estimates for the true distribution since the sample size is small. To hedge against distributional uncertainty, we propose a distributionally robust counterpart of the weighted $k$-NN problem defined in the previous subsection. Specifically, suppose each class $m$ is associated with a distributional uncertainty set $\mathcal{P}_m$, which will be specified shortly. Given $\mathcal{P}_1, \ldots, \mathcal{P}_M$, define the *worst-case risk* of a classifier $\pi$ as the worst-case total error probabilities

$$\max_{P_m \in \mathcal{P}_m, 1 \leq m \leq M} \Psi(\pi; P_1, \ldots, P_M),$$

where $\Psi$ is defined in (1).

We consider the following distributionally robust $k$-NN problem that finds the optimal weighted $k$-NN classifier minimizing the worst-case risk:

$$\min_{\substack{1 \leq k \leq n \\ w_m : \Xi \times \Xi \to \mathbb{R}_+ \\ 1 \leq m \leq M}} \max_{\substack{P_m \in \mathcal{P}_m \\ 1 \leq m \leq M}} \Psi(\pi^{\mathsf{knn}}(\cdot; k, w); P_1, \ldots, P_M). \tag{6}$$

Here the optimal solution $P_1^*, \ldots, P_M^*$ to the inner maximization problem is also called *least favorable distributions (LFD)* in statistics literature [21]. We summarize the architecture of the proposed distributionally robust $k$-NN framework in Figure 3, more details are provided in Section 4.

Now we describe the uncertainty set $\mathcal{P}_m$. First, since we are going to re-weight the training samples to build the classifier, we restrict the support of every distribution in $\mathcal{P}_m$ to $\widehat{\Xi}$, the set of empirical points. Second, the uncertainty set is data-driven, containing the empirical distribution $\widehat{P}_m$ and distributions surrounding its neighborhood. Third, to measure the closeness between distributions, we choose the Wasserstein metric of order 1 [40], defined as

$$\mathcal{W}(P, P') := \min_{\gamma} \mathbb{E}_{(\xi, \xi') \sim \gamma} \left[ c(\xi, \xi') \right]$$

for any two distributions $P$ and $P'$ on $\Xi$, where the minimization of $\gamma$ is taken over the set of all probability distributions on $\Xi \times \Xi$ with marginals $P$ and $P'$. The main advantage of using Wasserstein metric is that it takes account of the *geometry* of the feature space by incorporating the metric $c(\cdot, \cdot)$ in its definition. Given the empirical distribution $\widehat{P}_m$ for $m = 1, \ldots M$, we define

$$\mathcal{P}_m := \left\{ P_m \in \mathscr{P}(\widehat{\Xi}) : \mathcal{W}(P_m, \widehat{P}_m) \leq \vartheta_m \right\}, \tag{7}$$

where $\mathscr{P}(\widehat{\Xi})$ denotes the set of all probability distributions on $\widehat{\Xi}$; $\vartheta_m \geq 0$ specifies the size of the uncertainty set for the $m$-th class that specifies the amount of deviation we would like to control.

## 3 Theoretical Properties

In this section, we analyze the computational tractability and statistical properties of the proposed distributionally robust weighted $k$-NN classifier found in (6). All proofs are delegated to Appendix B.

### 3.1 Robust Classification

Observe that similar to (5), the formulation (6) is also an infinite-dimensional functional optimization. Let us first relate it to a relaxed robust classification problem, which turns out to be more tractable.

Consider the following minimax robust classification problem over all randomized classifiers $\pi$ (recalling $\Delta_M$ is the probability simplex in $\mathbb{R}_+^M$):

$$\min_{\pi : \Xi \to \Delta_M} \max_{P_m \in \mathcal{P}_m, 1 \leq m \leq M} \Psi(\pi; P_1, \ldots, P_M). \tag{8}$$

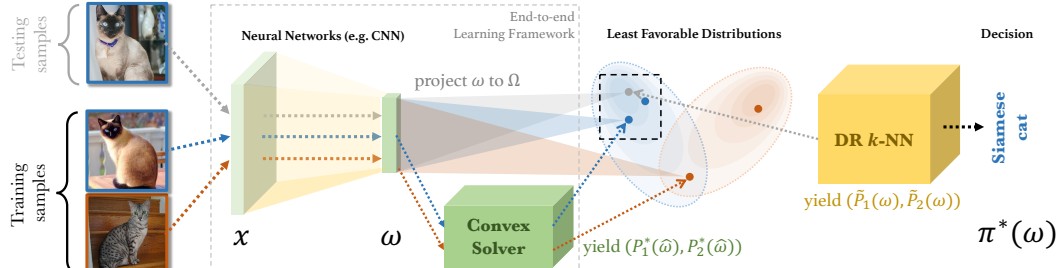

Figure 3: An overview of the end-to-end learning framework, which consists of two cohesive components: (1) an architecture that is able to produce feature embedding $\xi$ and least favorable distributions $P_m^*$ for training set; (2) an `Dr.k-NN` makes decisions for any unseen sample $\xi$ based on the estimated weight vector $\widetilde{p}_m(\xi)$ (probability mass on least favorable distributions).

Yet still, (8) is an infinite-dimensional functional optimization, since we are optimizing over the set of all randomized classifiers. We establish the following theorem stating a finite-dimensional convex programming reformulation for the problem (8).

**Theorem 1.** *For the uncertainty sets defined in* (7), *the least favorable distribution of problem* (8) *can be obtained by solving the following problem:*

$$
\begin{aligned}
\min_{\substack{p_1,\dots,p_M \in \mathbb{R}_+^n \\ \gamma_1,\dots,\gamma_M \in \mathbb{R}_+^{n \times n}}} \quad & \sum_{i=1}^n \max_{1 \le m \le M} p_m^i \\
\text{subject to} \quad & \sum_{i=1}^n \sum_{j=1}^n \gamma_m^{i,j} c(\xi^i, \xi^j) \le \vartheta_m, \\
& \sum_{i=1}^n \gamma_m^{i,j} = \widehat{P}_m(\xi^j), \quad \sum_{j=1}^n \gamma_m^{i,j} = p_m^i, \\
& \forall 1 \le i,j \le N,\ 1 \le m \le M.
\end{aligned}
\tag{9}
$$

The decision variable $\gamma_m \in \mathbb{R}_+^{n \times n}$ can be viewed as a joint distribution on $n$ empirical points with marginal distributions $\widehat{P}_m$ and $P_m$, represented by a vector $p_m \in \mathbb{R}_+^n$. The inequality constraint controls the Wasserstein distance between $P_m$ and $\widehat{P}_m$.

Below we give an intuitive explanation for the objective function in (9). Note that $\max_{1 \le m' \le M} p_{m'}^i - p_m^i$ measures the margin between the maximum likelihood of $\xi^i$ among all classes and the likelihood of the $m$-th class. Thus, the objective in (9) can be equivalently rewritten as minimization of total margin:

$$
\sum_{i=1}^n \sum_{m=1}^M \left( \max_{1 \le m' \le M} p_{m'}^i - p_m^i \right).
$$

When $M = 2$, the total margin reduces to the total variation distance. Also, let $y_m^i \in \{0,1\}$ be the class indicator variable of sample $\xi^i$, observe that

$$
\sum_{i=1}^n \max_{1 \le m \le M} p_m^i = \lim_{t \to \infty} \left( \sum_{i=1}^n \sum_{m=1}^m y_m^i p_m^i - \frac{1}{t} \sum_{i=1}^n \sum_{m=1}^M y_m^i \log \frac{\exp(t p_m^i)}{\sum_{m=1}^M \exp(t p_m^i)} \right),
$$

where the second term on the right side represents the cross-entropy (or negative log-likelihood).

Therefore, problem (9) perturbs $(\widehat{P}_1, \dots, \widehat{P}_M)$ to LFDs $(P_1^*, \dots, P_M^*)$ so as to minimize the total margin as well as an upper bound on cross-entropy of LFDs; the smaller the margin (or cross-entropy) is, the more similar between classes and thus the harder to distinguish among them.

### 3.2 Expressiveness of Weighted $k$-NN

In this subsection we study the expressive power of the class of weighted $k$-NN classifiers

$$
\{\pi^{\mathsf{knn}}(\cdot; k, w) \colon 1 \le k \le n, w_m \colon \Xi \times \Xi \to \mathbb{R}_+, 1 \le m \le M\}
$$

defined in Section 2.1.

The following theorem establishes the equivalence between the original problem (6) and the relaxed robust classification problem (8) studied in Section 3.1.

**Theorem 2.** *For the uncertainty set defined in* (7), *formulations* (6) *and* (8) *have identical optimal values. In addition, there exists optimal solutions of* (6) *and* (8) *that share common LFDs that are optimal to* (9).

Theorem 2 implies that the set of weighted $k$-NN classifiers is *exhaustive*, in the sense that it achieves the same optimal robust risk as optimizing over the set of all randomized classifiers. In our proof, we show that the *weighted 1-NN classifier*, with weights equal to the LFDs of (9), is an optimal solution to (6). Therefore, instead of solving (6) directly, by Theorem 1, we can solve the convex program (9) for the LFDs, based on which we construct a robust $k$-NN classifier. This justifies the `Dr.k-NN` algorithm to be described in Section 4.

### 3.3 Lipschitz Regularization

Next, we discuss the connection between the proposed distributional robust $k$-NN framework in (6), (8) and an equivalent Lipschitz regularization problem.

Using duality for Wasserstein DRO [17], problem (8) is equivalent to

$$\min_{\substack{\pi:\widehat{\Xi}\to\Delta_M \\ \lambda_m\geq 0, m=1,\dots M}} \left\{ \sum_{m=1}^{M} \lambda_m\vartheta_m + \mathbb{E}_{\hat{\xi}\sim\widehat{P}_m} \left[ \max_{\xi\in\widehat{\Xi}} \left\{ 1 - \pi_m(\xi) - \lambda_m c(\xi,\hat{\xi}) \right\} \right] \right\}. \tag{10}$$

Then by [16], this problem is upper bounded by the following Lipschitz regularized classification problem

$$\min_{\pi:\widehat{\Xi}\to\Delta_M} \sum_{m=1}^{M} \left\{ \mathbb{E}_{\xi\sim\widehat{P}_m} [1 - \pi_m(\xi)] + \vartheta_m\|\pi_m\|_{\text{Lip}} \right\}, \tag{11}$$

where

$$\|\pi_m\|_{\text{Lip}} := \max_{\xi,\tilde{\xi}\in\widehat{\Xi},\,\xi\neq\tilde{\xi}} \frac{|\pi_m(\tilde{\xi}) - \pi_m(\xi)|}{c(\tilde{\xi},\xi)}$$

is the Lipschitz norm of the function $\pi_m$ for each $m = 1,\dots,M$. Perhaps surprisingly, the next result shows that (10) and (11) are actually equivalent (thus by Theorem 2, are both equivalent to (6)), despite that the loss function does not satisfy existing criteria ensuring the equivalence [13, 36, 16].

**Theorem 3.** *Formulations* (10) *and* (11) *are equivalent, and there exists an optimizer* $\pi^*$ *of* (11) *satisfying* $\|\pi_m^*\|_{\text{Lip}} = \lambda_m^*$, $m = 1,\dots,M$, *where* $(\lambda_m^*)_m$ *is the optimizer of* (10) *when* $\pi = \pi^*$.

The theorem is proved by using $c$-transform [40] to show that any optimizer $\pi_m^*$ can be modified into a $\lambda_m^*$-Lipschitz classifier while maintaining the optimality. Note that Lipschitz classifiers are known to enjoy good generalization property [42, 20], thus the equivalence provides promising insights for the generalization property of the robust classifier.

## 4 Proposed Algorithm `Dr.k-NN`

In this section, we present the Distributional robust $k$-Nearest Neighbor (`Dr.k-NN`) algorithm, which is a direct consequence of the theoretical justifications in Section 3.

### 4.1 `Dr.k-NN` algorithm

Based on Theorem 2, our algorithm contains two steps.

**Step 1.** [*Sample re-weighting*] For each class $m$, re-weight $n$ samples using a distribution $P_m^*$, where $(P_1^*,\dots,P_M^*)$ is the $p$-component of the minimizer of (9).

**Step 2.** [*k-NN*] Given a query point $\xi$, ordering the training samples according to their distance to $\xi$: $c(\xi,\xi^{\tau_1^S(\xi)}) \leq c(\xi,\xi^{\tau_2^S(\xi)}) \leq \cdots \leq c(\xi,\xi^{\tau_n^S(\xi)})$. Compute the weighted $k$-NN votes, define

$$\widetilde{p}_m(\xi) := \frac{1}{k}\sum_{i=1}^{k} P_m^*(\xi^{\tau_i^S(\xi)}), \ m = 1,\dots,M. \tag{12}$$

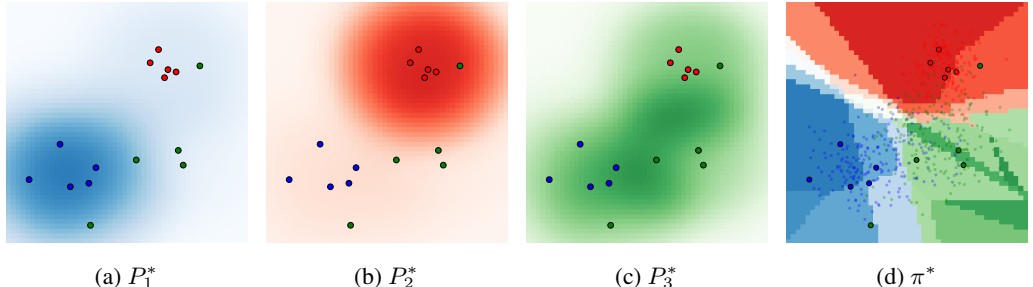

| (a) $P_1^*$ | (b) $P_2^*$ | (c) $P_3^*$ | (d) $\pi^*$ |

Figure 4: An example of the weights $P_1^*, P_2^*, P_3^*$ yielding from (9) and the corresponding results of Dr.k-NN using a small subset of MNIST (digit 4 (red), 6 (blue), 9 (green) and $k = 5$). Raw samples are projected on a 2D feature space ($d = 2$), with the color indicating their true class-membership. In (a)(b)(c), shaded areas indicate the kernel smoothing of $P_1^*, P_2^*, P_3^*$ defined in (13). In (d), big dots represent the training points and small dots represent the query points, and their color depth suggests how likely the sample is being classified into the true category.

Table 1: Comparison of classification accuracy in the few-training-sample setting

| Methods | MNIST | | | | mini ImageNet | | | | CIFAR-10 | | | | Omniglot | | | | Lung Cancer | | COVID-19 CT | |
|---|---|---|---|---|---|---|---|---|---|---|---|---|---|---|---|---|---|---|---|---|
| | $M=2$ | | $M=5$ | | $M=2$ | | $M=5$ | | $M=2$ | | $M=5$ | | $M=2$ | | $M=5$ | | $M=3$ | | $M=2$ | |
| | $K=5$ | $K=10$ | $K=5$ | $K=10$ | $K=5$ | $K=10$ | $K=5$ | $K=10$ | $K=5$ | $K=10$ | $K=5$ | $K=10$ | $K=5$ | $K=10$ | $K=5$ | $K=10$ | $K=5$ | $K=8$ | $K=5$ | $K=10$ |
| PCA+$k$-NN | 0.801 | 0.872 | 0.614 | 0.678 | 0.578 | 0.667 | 0.268 | 0.277 | 0.687 | 0.711 | 0.262 | 0.270 | 0.597 | 0.638 | 0.309 | 0.358 | 0.617 | 0.647 | 0.658 | 0.719 |
| SVD+$k$-NN | 0.749 | 0.790 | 0.524 | 0.567 | 0.587 | 0.675 | 0.268 | 0.283 | 0.680 | 0.701 | 0.259 | 0.266 | 0.591 | 0.618 | 0.305 | 0.413 | 0.624 | 0.648 | 0.646 | 0.715 |
| NCA+$k$-NN | 0.602 | 0.640 | 0.340 | 0.355 | 0.547 | 0.578 | 0.245 | 0.258 | 0.597 | 0.616 | 0.232 | 0.236 | 0.549 | 0.574 | 0.267 | 0.346 | 0.575 | 0.582 | 0.612 | 0.624 |
| Matching Net | 0.732 | 0.830 | 0.625 | 0.732 | 0.687 | 0.703 | 0.286 | **0.360** | 0.632 | 0.641 | 0.241 | 0.247 | 0.735 | 0.769 | 0.412 | 0.433 | 0.621 | 0.635 | 0.715 | 0.732 |
| Prototypical Net | 0.742 | 0.842 | 0.671 | 0.759 | 0.710 | 0.725 | 0.296 | 0.348 | 0.651 | 0.664 | 0.254 | 0.259 | **0.769** | 0.836 | 0.448 | 0.532 | 0.632 | 0.644 | **0.729** | **0.744** |
| MetaOptNet | 0.725 | 0.843 | 0.658 | 0.790 | 0.732 | 0.741 | 0.255 | 0.363 | 0.702 | 0.713 | 0.257 | 0.298 | 0.742 | 0.755 | 0.412 | 0.453 | 0.638 | 0.642 | 0.713 | 0.739 |
| Feature embedding + $k$-NN | 0.792 | 0.798 | 0.546 | 0.551 | 0.738 | 0.742 | **0.490** | 0.486 | 0.689 | 0.691 | **0.492** | **0.494** | 0.725 | 0.751 | 0.445 | 0.495 | 0.664 | 0.691 | 0.701 | 0.710 |
| Kernel Smoothing | 0.777 | 0.873 | 0.559 | 0.579 | 0.593 | 0.601 | 0.272 | 0.278 | 0.642 | 0.661 | 0.272 | 0.282 | 0.520 | 0.565 | 0.240 | 0.285 | 0.367 | 0.370 | 0.582 | 0.604 |
| **Truncated Dr.k-NN** | 0.815 | 0.926 | **0.742** | 0.825 | 0.746 | 0.753 | 0.295 | 0.340 | 0.703 | **0.719** | 0.297 | 0.305 | 0.755 | 0.825 | 0.425 | **0.542** | 0.652 | **0.693** | 0.722 | 0.741 |
| **Dr.k-NN** | **0.838** | **0.959** | 0.746 | **0.831** | **0.752** | **0.786** | 0.306 | 0.358 | **0.707** | 0.728 | 0.309 | 0.311 | 0.765 | **0.850** | **0.465** | 0.580 | **0.667** | 0.704 | 0.734 | 0.752 |

Decide the class for a query feature point $\xi$ as $\arg\max_{1 \leq m \leq M} \widetilde{p}_m(\xi)$, where the tie is broken according to the rule (4).

Figure 4 gives an illustration showing the probabilistic weights $(P_1^*, P_2^*, P_3^*)$ for three classes and its corresponding decision boundary yielding from the weighted $k$-NN.

For the sake of completeness, we also extend our algorithm to non-few-training-sample setting, which is referred to as truncated Dr.k-NN. The key idea is to keep the training samples that are important in deciding the decision boundary based on the maximum entropy principle [11]. This can be particularly useful for the general classification problem with an arbitrary size of training set. An illustration (Figure 6) and more details can be found in Appendix C.

## 4.2 Joint learning framework

In this section, we propose a framework that jointly learns the feature mapping and the robust classifier. Let the feature mapping $\phi(;\theta)$ be a neural network parameterized by $\theta$ whose input is a batch of training samples (Figure 3), and then compose it with an optimization layer that packs the convex problem (9) as an output layer that outputs the LFDs of (8). The optimization layer is adopted from differentiable optimization [4, 2], in which the optimization problem is integrated as an individual layer in an end-to-end trainable deep networks and the solution of the problem can be backpropagated through neural networks.

To apply the mini-batch stochastic gradient descent, we need to ensure that each batch comprises of multiple "mini-sets", one for each class, containing at least one training sample from each class fed into the convex optimization layer. In light of (9), the objective of our joint learning framework is $\min_\theta J(\theta; P_1^*, \ldots, P_M^*)$, where

$$J(\theta; P_1^*, \ldots, P_M^*) \coloneqq \sum_{i=1}^n \max_{1 \leq m \leq M} P_m^*(\phi(x^i; \theta)),$$

and $\{P_m^*(\phi(\cdot; \theta))\}_{1 \leq m \leq M}$ are the LFDs generated by the convex solver defined in (9) given input variables $\{\xi^i = \phi(x^i; \theta)\}_{1 \leq i \leq n}$. The algorithm is summarized in Algorithm 1 (Appendix A).

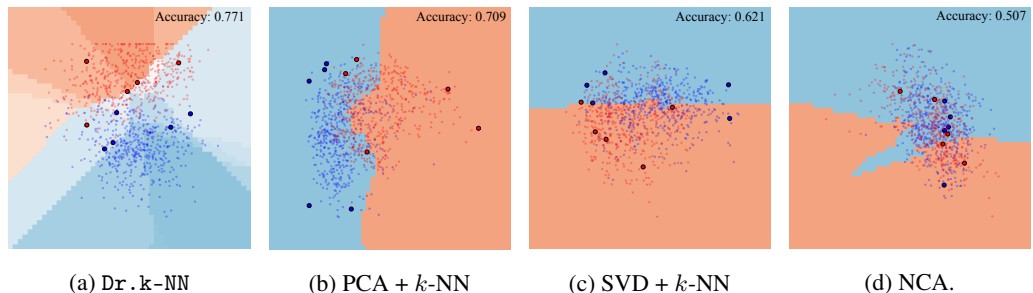

| (a) Dr.k-NN | (b) PCA + $k$-NN | (c) SVD + $k$-NN | (d) NCA. |

Figure 5: A comparison of the learned feature spaces and the corresponding decision boundaries. There are 10 training samples from two categories of MNIST identified as large dots and 1,000 query samples identified as small dots. The color of dots shows their true categories. The color of the region shows the decisions made by corresponding methods.

## 5 Experiments

In this section, we evaluate our method and eight alternative approaches on four commonly-used image data sets: MNIST [28], CIFAR-10 [25], Omniglot [27], and present a set of comprehensive numerical examples.

We also test our method on two medical diagnosis data sets: Lung Cancer [12], and COVID-19 CT [44], where very few data samples are available for study, due to privacy concerns and high costs associated with harvesting data. Specifically, Lung Cancer data record 56 attributes for only 32 patients who have been diagnosed with three types of pathological lung cancers; COVID-19 Computed Tomography (CT) data contain 349 COVID-19 CT images from 216 patients and 463 non-COVID-19 CTs. Here, we present a few examples of COVID-19 CT images and non-COVID-19 CT images in Appendix E.

**Experiment set-up.** We compare our method including Dr.k-NN and its truncated version with the following baselines: (1) $k$-NN based methods with different dimension reduction techniques, including Principal Component Analysis (PCA+$k$-NN), Singular Value Decomposition (SVD+$k$-NN), Neighbourhood Components Analysis (NCA+$k$-NN) [19], and feature embeddings generated by Dr.$k$-NN (Feature embedding + $k$-NN) as a sanity check; (2) matching networks [41]; (3) prototypical networks [39]; (4) MetaOptNet [29]. To make these methods comparable, we adopt the same naive neural network with a single CNN layer on matching network, prototypical network, and our model, respectively, where the kernel size is 3, the stride is 1 and the width of the output layer is $d = 400$.

In our experiments, we focus on an $M$-class $K$-sample ($K$ training samples for each class) learning task. To generate the training data set, we randomly select $M$ classes and for each class we take $K$ random samples. So our training data set contains $MK$ samples overall. We then aim to classify a disjoint batch of unseen samples into one of these $M$ classes. Thus random performance on this task stands at $1/M$. We test the average performance of different methods using $1,000$ unseen samples from the same $M$ classes. To obtain reliable results, we repeat each test 10 times and calculate the average accuracy.

Other experimental configurations are described as follows: The Adam optimizer [23] is adopted for all experiments conducted in this paper, where learning rate is $10^{-2}$. The mini-batch size is 32. The hyper-parameter $\vartheta_m$ is chosen by cross-validation, which varies from application to application. The differentiable convex optimization layer we adopt is from [2]. To make all approaches comparable, we use the same network structure in matching network, prototypical network, and MetaOptNet as we described above. We use the Euclidean distance $c(\xi, \xi') = \|\xi - \xi'\|_2$ throughout our experiment. All experiments are performed on Google Colaboratory (Pro version) with 12GB RAM and dual-core Intel processors, which speed up to 2.3 GHz (without GPU).

**Results.** We present the average test accuracy in Table 1 for the unseen samples with different $M = 2, 5$ and $K = 5, 10$ on small subsets of MNIST, *mini* ImageNet, CIFAR-10, Omniglot, and with $M = 3$ and $K = 5, 8$ on lung cancer data. Note that random performance for two-class and five-class classifications are $0.5$ and $0.2$, respectively. The figures in Table 1 show that Dr.k-NN ($k = 5$) outperforms other baselines in terms of the average test accuracy on all data sets. We note

that 95% confidence interval of our method's performance on all the data sets is smaller than 0.08. The truncated `Dr.k-NN` also yields competitive results using only 20% training samples ($\tau = 0.9$), compared to standard `Dr.k-NN`.

To confirm that the proposed learning framework will affect the distribution of hidden representation of data points, we show the training and query samples in a 2D feature space and the corresponding decision boundary in Figure 5. It turns out our framework finds a better feature representation in the 2D space with a smooth decision boundary and a reasonable decision confidence map (indicated by the color depth in Figure 5 (a)).

**Comparison to kernel smoothing.**  We also compare with an approach using kernel-smoothing of the LFDs (in contrast to using $k$-NN) for performing classification. Consider a Gaussian kernel $\kappa(x) = |H_h|^{-1/2}\kappa(H_h^{-1/2}x)$, where $H_h = hI_p$ is the isotropical kernel with bandwidth $h$. Then we replace $\widetilde{p}_m(\xi)$ in Step 2 of `Dr.k-NN` with the following

$$\widetilde{p}_m(\xi) := \sum_{i=1}^{n} P_m^*(\xi^i)\kappa(\xi - \xi^i), \ \forall \xi, \ m = 1, \dots, M. \tag{13}$$

We evaluate both methods on a subset of MNIST, which contains 1,000 testing samples (small dots) and 20 training samples (large dots) from two categories (indexed by blue and red, respectively).

As shown in Figure 7 and Figure 8 (Appendix D), our experimental results have shown the importance of using $k$-NN in our proposed algorithm, where the `Dr.kNN` significantly outperforms the parallel version using kernel smoothing (even after the kernel bandwidth being optimized). We find that the performance when using kernel smoothing (13) heavily depends on selecting an appropriate kernel bandwidth $h$ as illustrated by Figure 8.

Moreover, the best kernel bandwidth may vary from one dataset to another. Therefore, the cross-validation is required to be carried out to find the best kernel bandwidth in practice, which is quite time-consuming. In contrast, choosing the hyper-parameter $k$ is an easy task, since we only have limited choices of $k$ in few-training-sample scenario and the performance of `Dr.k-NN` is insensitive to the choices of $k$ (see Figure 7).

## 6    Conclusion

We propose a distributionally robust $k$-NN classifier (`Dr.k-NN`) for tackling the multi-class classification problem with few training samples. To make a decision, each neighboring sample is weighted according to least favorable distributions resulting from a distributionally robust problem. As shown in the theoretical results and demonstrated by experiments, our methods achieve outstanding performance in classification accuracy compared with other baselines using minimal resources. The robust classifier layer (9) serves an alternative to the usual softmax layer in a neural network for classification, and we believe it is promising for other machine learning tasks.

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
