# A  Learning Algorithm

---

**Algorithm 1:** Learning algorithm for `Dr.k-NN`

---

**Input:** $S_m := \{(x^i, y^i) : y^i = m, \forall i\} \subset S, \ m = 1, \dots, M$;
**Output:** The feature mapping $\phi(\cdot; \theta)$ and the LFD $P_1^*, \dots, P_M^*$ supported on training samples;
**Initialization:** $\theta_0$ is randomly initialized; $n' < n$ is the size of "mini-set"; $t = 0$;
**while** $t < T$ **do**
    **for** *number of mini-sets* **do**
        Randomly generate $M$ integers $n_1, \dots, n_M$ such that $\sum_{m=1}^{M} n_m = n', n_m > 0, \forall m$;
        Initialize two ordered sets $\widehat{\Xi} = \emptyset, \widehat{P} = \emptyset$;
        **for** $m \in \{1, \dots, M\}$ **do**
            $\mathcal{X}_m \leftarrow$ Randomly sample $n_m$ points from $S_m$;
            $\widehat{\Xi}_m \leftarrow \{\xi := \phi(x; \theta_t) : x \in \mathcal{X}_m\}$;
            $\widehat{P}_m \leftarrow \frac{1}{n_m} \sum_{i=1}^{n_m} \delta_{\xi_m^i}, \ \xi_m^i \in \widehat{\Xi}_m$;
            $\widehat{\Xi} \leftarrow \widehat{\Xi} \cup \widehat{\Xi}_m; \widehat{P} \leftarrow \widehat{P} \cup \widehat{P}_m$;
        **end**
        Update the probability mass of LFDs $P_1^*, \dots, P_M^*$ on $\widehat{\Xi}$ by solving (9) given $\widehat{\Xi}, \widehat{P}$;
    **end**
    $\theta_{t+1} \leftarrow \theta_t - \alpha \nabla J(\theta_t; P_1^*, \dots, P_M^*)$, where $\alpha$ is the learning rate;
    $t \leftarrow t + 1$;
**end**

---

# B  Proofs for Section 3

## B.1  Proof of Theorem 1

The proof of Theorem 1 is based on the following two lemmas.

**Lemma 1.** *Fix probability distributions* $P_1, \dots, P_M \in \mathscr{P}(\widehat{\Xi})$, *where* $\widehat{\Xi} = \{\xi^1, \dots, \xi^n\}$. *Then*

$$\psi(P_1, \dots, P_M) := \min_{\pi:\widehat{\Xi} \to \Delta_M} \Psi(\pi; P_1, \dots, P_M) = M - \sum_{i=1}^{n} \max_{1 \le m \le M} P_m(\xi^i).$$

*Furthermore, the optimal classifier* $\pi^*$ *satisfies that for any* $\xi^i \in \widehat{\Xi}$,

$$\left\{ m : \pi_m^*(\xi^i) > 0, 1 \le m \le M \right\} \subset \arg\max_{1 \le m \le M} \frac{P_m(\xi^i)}{\sum_{m=1}^{M} P_m(\xi^i)}.$$

This lemma gives a closed-form expression for the risk of the optimal classifier if $P_1, \dots, P_M$ are known, and shows that the optimal decision $\pi^*$ accepts the class with the maximum likelihood. Moreover, when there is a tie (i.e., the set $\arg\max_{1 \le m \le M} P_m(\xi)$ is not singleton), the optimal decision $\pi^*$ can break the tie arbitrarily.

*Proof of Lemma 1.* We here prove a more general result for an arbitrary sample space $\Xi$. Note that each $P_m, 1 \le m \le M$, is absolutely continuous with respect to $P_1 + \cdots + P_M$, hence the Radon-Nikodym derivative $\frac{dP_m}{d(P_1 + \cdots + P_M)}$ exists. Using the interchangeability principle [37] that enables us to exchange the minimization and integration, we have

$$\min_{\pi:\Xi \to \Delta_M} \Psi(\pi; P_1, \dots, P_M) = \min_{\pi:\Xi \to \Delta_M} \int_{\Xi} \Big[ \sum_{m=1}^{M} (1 - \pi_m(\xi)) \frac{dP_m}{d(P_1 + \cdots + P_M)}(\xi) \Big] d(P_1 + \cdots + P_M)$$

$$= \int_{\Xi} \min_{\pi \in \Delta_M} \Big[ \sum_{m=1}^{M} (1 - \pi_m) \frac{dP_m}{d(P_1 + \cdots + P_M)}(\xi) \Big] d(P_1 + \cdots + P_M)$$

$$= \int_{\Xi} \Big[ 1 - \max_{1 \le m \le M} \frac{dP_m}{d(P_1 + \cdots + P_M)}(\xi) \Big] d(P_1 + \cdots + P_M),$$

$$(14)$$

where the first equality is obtained by plugging in the definition of $\Psi$ in (1); the second equality is due the interchangeability principle; and the last equality holds because for any $\xi$, the inner minimization attains its minimum at one of the vertices of $\Delta_M$. More specifically, note that for each $\xi$, the objective of the inner minimization problem equals to $1 - \sum_{m=1}^{M} \pi_m \frac{dP_m}{d(P_1 + \cdots + P_M)}(\xi)$. Under the constraint that $\pi \in \Delta_M$, i.e., $\sum_{m=1}^{M} \pi_m = 1$, we have:

$$\sum_{m=1}^{M} \pi_m \frac{dP_m}{d(P_1 + \cdots + P_M)}(\xi) \leq \max_{1 \leq m \leq M} \frac{dP_m}{d(P_1 + \cdots + P_M)}(\xi),$$

and the equality holds when $\pi$ are chosen such that

$$\{m : \pi_m > 0, 1 \leq m \leq M\} \subset \arg\max_{1 \leq m \leq M} \frac{dP_m(\xi)}{\sum_{m=1}^{M} dP_m(\xi)}.$$

If there is a single maximum in $\{\frac{dP_m}{d(P_1 + \cdots + P_M)}(\xi), 1 \leq m \leq M\}$, say at index $m^*$, then this simply implies that the optimal $\pi$ is chosen as $\pi_{m^*} = 1$ and $\pi_m = 0$ for $m \neq m^*$.

If we substitute $\Xi$ with the empirical support $\widehat{\Xi}$, the above formulation in Equation (14) translates into

$$\min_{\pi:\widehat{\Xi} \to \Delta_M} \Psi(\pi; P_1, \ldots, P_M) = M - \sum_{i=1}^{n} \max_{1 \leq m \leq M} P_m(\xi^i),$$

therefore the lemma is proved. $\qquad\square$

**Lemma 2.** *For the uncertainty sets defined in (7), the problem* $\max_{P_m \in \mathcal{P}_m, 1 \leq m \leq M} \psi(P_1, \ldots, P_M)$ *is equivalent to (9).*

*Proof of Lemma 2.* Recall that the Wasserstein metric of order 1 is defined as

$$\mathcal{W}(P, P') := \min_{\gamma} \mathbb{E}_{(\xi, \xi') \sim \gamma} [c(\xi, \xi')]$$

for any two distributions $P$ and $P'$ on $\Xi$, where the minimization of $\gamma$ is taken over the set of all probability distributions on $\Xi \times \Xi$ with marginals $P$ and $P'$, i.e., the set

$$\left\{ \gamma \in \mathscr{P}(\Xi \times \Xi) : \int_{\Xi} \gamma(\xi, \xi') d\xi' = P(\xi), \int_{\Xi} \gamma(\xi, \xi') d\xi = P'(\xi'), \forall \xi, \xi' \in \Xi \right\},$$

where $\mathscr{P}(\Xi \times \Xi)$ denotes the joint probability distributions on $\Xi \times \Xi$. Therefore, the Wasserstein metric $\mathcal{W}(P, P')$ can be rewritten as

$$\min_{\gamma} \left\{ \int_{\Xi \times \Xi} c(\xi, \xi') \gamma(\xi, \xi') d\xi d\xi' : \int_{\Xi} \gamma(\xi, \xi') d\xi' = P(\xi), \int_{\Xi} \gamma(\xi, \xi') d\xi = P'(\xi'), \forall \xi, \xi' \in \Xi \right\}$$

By the definition of uncertainty sets in (7) which contains discrete distributions supported on $\widehat{\Xi}$, we can introduce additional variables $\gamma_m \in \mathbb{R}_+^{n \times n}$ which represents the distribution on $\widehat{\Xi} \times \widehat{\Xi}$, with marginals $P_m \in \mathcal{P}_m$ and $\widehat{P}_m$, for $1 \leq m \leq M$. For any $\xi^i, \xi^j \in \widehat{\Xi}$, let $\gamma_m^{i,j}$ denotes $\gamma_m(\xi^i, \xi^j)$ for simplicity. Thus the objective function in the above reformualtion of $\mathcal{W}(P_m, \widehat{P}_m)$ is $\sum_{i=1}^{n} \sum_{j=1}^{n} \gamma_m^{i,j} c(\xi^i, \xi^j)$. The constraints $\mathcal{W}(P_m, \widehat{P}_m) \leq \vartheta_m$ in (7) can be rewritten using $\gamma_m$ as

$$\sum_{i=1}^{n} \sum_{j=1}^{n} \gamma_m^{i,j} c(\xi^i, \xi^j) \leq \vartheta_m, \quad m = 1, \ldots, M.$$

Furthermore, the marginal distribution constraint of $\gamma_m$ reads:

$$\sum_{i=1}^{n} \gamma_m^{i,j} = \widehat{P}_m(\xi^j), \quad \sum_{j=1}^{n} \gamma_m^{i,j} = P_m(\xi^i), \quad m = 1, \ldots, M.$$

Thereby the problem $\max_{P_m \in \mathcal{P}_m, 1 \leq m \leq M} \psi(P_1, \ldots, P_M)$ is equivalent to the convex optimization formulation in (9). $\qquad\square$

*Proof to Theorem 1.* By Lemmas 1 and 2, we have

$$\max_{P_m \in \mathcal{P}_m, 1 \le m \le M} \min_{\pi : \widehat{\Xi} \to \Delta_M} \Psi(\pi; P_1, \ldots, P_M) = \max_{P_m \in \mathcal{P}_m, 1 \le m \le M} \psi(P_1, \ldots, P_M) = (9).$$

To prove Theorem 1, it remains to verify the validity of exchanging $\max$ and $\min$. We identify $\pi$ as $(\pi^1, \ldots, \pi^n)$, where $\pi^i \in \mathbb{R}_+^M$ satisfies $\sum_{m=1}^M \pi_m^i = 1$. Similar to the proof of Lemma 2, $P_m$, $1 \le m \le M$, can also be identified as a vector in $\mathbb{R}^n$. Note that the objective function $\Psi(\pi; P_1, \ldots, P_M)$ is linear in $(\pi^1, \ldots, \pi^n)$ and concave in $(P_1, \ldots, P_M)$, and the Slater condition holds. Hence applying convex programming duality we can exchange $\max$ and $\min$ and thus the result follows. It is worth mentioning that the optimal solution $\pi^*$ and corresponding LFDs $P_1^*, \ldots, P_M^*$ always exist since they are solutions to a saddle point problem. □

## B.2 Proof of Theorem 2

*Proof of Theorem 2.* On the one hand, since $\pi^{\mathsf{knn}}(\cdot; k, w)$ can be regarded as a special case of the general classifier $\pi : \Xi \to \Delta_M$, it holds that

$$\min_{\substack{w_m : \Xi \times \Xi \to \mathbb{R}_+, \, 1 \le m \le M \\ 1 \le k \le n}} \max_{P_m \in \mathcal{P}_m, 1 \le m \le M} \sum_{m=1}^M \mathbb{E}_{\xi_m \sim P_m}[1 - \pi_m^{\mathsf{knn}}(\xi_m; k, w)]$$

$$\ge \min_{\pi : \widehat{\Xi} \to \Delta_M} \max_{P_m \in \mathcal{P}_m, 1 \le m \le M} \sum_{m=1}^M \mathbb{E}_{\xi_m \sim P_m}[1 - \pi_m(\xi_m)].$$

On the other hand, by Lemma 2, there exists an optimal solution to the minimax problem (8), denoted as $(P_1^*, \ldots, P_M^*)$, and the optimal classifier $\pi^*$ as given in Lemma 1. Note that there exists $1 \le k^* \le n$ and weight functions $w_1^*, \ldots, w_M^*$ such that

$$\pi_m^{\mathsf{knn}}(\xi; k^*, w^*) = \pi_m^*(\xi), \quad \forall \xi \in \widehat{\Xi}, \tag{15}$$

for example, by taking $k^* = 1$ and $w_m^* = P_m^*$, $1 \le m \le M$. This implies that

$$\min_{\substack{w_m : \Xi \times \Xi \to \mathbb{R}_+, \, 1 \le m \le M \\ 1 \le k \le K}} \max_{P_m \in \mathcal{P}_m, 1 \le m \le M} \sum_{m=1}^M \mathbb{E}_{\xi_m \sim P_m}[1 - \pi_m^{\mathsf{knn}}(\xi_m; k, w)]$$

$$\le \max_{P_m \in \mathcal{P}_m, 1 \le m \le M} \sum_{m=1}^M \mathbb{E}_{\xi_m \sim P_m}[1 - \pi_m^{\mathsf{knn}}(\xi_m; k^*, w^*)]$$

$$= \max_{P_m \in \mathcal{P}_m, 1 \le m \le M} \sum_{m=1}^M \mathbb{E}_{\xi_m \sim P_m}[1 - \pi_m^*(\xi_m)]$$

$$= \min_{\pi : \widehat{\Xi} \to \Delta_M} \max_{P_m \in \mathcal{P}_m, 1 \le m \le M} \sum_{m=1}^M \mathbb{E}_{\xi_m \sim P_m}[1 - \pi_m(\xi_m)].$$

Thereby we have shown that formulations (6) and (8) have identical optimal values. Moreover, by the strong duality results in Theorem 1, we know $(\pi^*; P_1^*, \ldots, P_M^*)$ is the saddle point for the formulation (8), and by the above arguments, we see that $(\pi^{\mathsf{knn}}; P_1^*, \ldots, P_M^*)$ leads to the same optimal value for formulation (6) as $(\pi^*; P_1^*, \ldots, P_M^*)$ for formulation (8). Therefore, we show that $\pi^{\mathsf{knn}}$ is indeed the optimal solution to (6). □

## B.3 Proof of Theorem 3

*Proof of Theorem 3.* We first show the equivalence between the Lipschitz regularized problem (11) and the minimax problem (8). Denote by $v_{\mathsf{Lip}}$ the optimal value of (11) and $v_{\mathsf{dual}}$ the optimal value of (10).

Observe that if $\|\pi_m\|_{\mathsf{Lip}} \le \lambda_m$, then

$$\max_{\xi \in \widehat{\Xi}} \left\{ 1 - \pi_m(\xi) - \lambda_m c(\xi, \widehat{\xi}) \right\} = 1 - \pi_m(\hat{\xi}), \; \forall \hat{\xi} \in \widehat{\Xi}.$$

Therefore, we have

$$v_{\text{dual}} \leq \min_{\pi: \widehat{\Xi} \to \Delta_M, \ (\lambda_m)_{1 \leq m \leq M} \geq 0, \ \|\pi_m\|_{\text{Lip}} \leq \lambda_m} \left\{ \sum_{m=1}^{M} \lambda_m \vartheta_m + \mathbb{E}_{\hat{\xi} \sim \widehat{P}_m} \left[ \max_{\xi \in \widehat{\Xi}} \left\{ 1 - \pi_m(\xi) - \lambda_m c(\xi, \hat{\xi}) \right\} \right] \right\}$$

$$= \min_{\pi: \widehat{\Xi} \to \Delta_M} \left\{ \sum_{m=1}^{M} \|\pi_m\|_{\text{Lip}} \vartheta_m + \mathbb{E}_{\hat{\xi} \sim \widehat{P}_m} \left[ 1 - \pi_m(\hat{\xi}) \right] \right\}$$

$$= v_{\text{Lip}}.$$

If we can show $v_{\text{dual}} \geq v_{\text{Lip}}$, then we prove the equivalence between (11) and (10), thus the equivalence between (11) and (8).

Let $(\pi^*; \lambda_1^*, \ldots, \lambda_M^*)$ be a dual minimizer of problem (10), whose existence is ensured by [17].

Define

$$\phi_m(\xi) := \max_{\tilde{\xi} \in \widehat{\Xi}} \left\{ 1 - \pi_m^*(\tilde{\xi}) - \lambda_m^* c(\tilde{\xi}, \xi) \right\}, \quad m = 1, \ldots, M.$$

Then it follows that

$$\pi_m^*(\tilde{\xi}) \geq 1 - \lambda_m^* c(\tilde{\xi}, \xi) - \phi_m(\xi), \ \forall \xi, \tilde{\xi} \in \widehat{\Xi}, \ m = 1, \ldots, M.$$

Define

$$\tilde{\pi}_m(\tilde{\xi}) := \max_{\xi \in \widehat{\Xi}} \{ 1 - \lambda_m^* c(\xi, \tilde{\xi}) - \phi_m(\xi) \}, \quad m = 1, \ldots, M.$$

Then by definition, $\|\tilde{\pi}_m\|_{\text{Lip}} \leq \lambda_m^*$. Indeed, for any $\xi, \tilde{\xi} \in \widehat{\Xi}$, there exists a $\xi_0 \in \arg\max_{\hat{\xi} \in \widehat{\Xi}} \{ 1 - \lambda_m^* c(\hat{\xi}, \xi) - \phi_m(\hat{\xi}) \}$ such that:

$$\begin{aligned} \tilde{\pi}_m(\xi) - \tilde{\pi}_m(\tilde{\xi}) &= 1 - \lambda_m^* c(\xi_0, \xi) - \phi_m(\xi_0) - \tilde{\pi}_m(\tilde{\xi}) \\ &\leq [1 - \lambda_m^* c(\xi_0, \xi) - \phi_m(\xi_0)] - [1 - \lambda_m^* c(\xi_0, \tilde{\xi}) - \phi_m(\xi_0)] \\ &= \lambda_m^* c(\xi_0, \tilde{\xi}) - \lambda_m^* c(\xi_0, \xi) \\ &\leq \lambda_m^* c(\xi, \tilde{\xi}). \end{aligned}$$

Furthermore, since $\tilde{\pi}_m(\tilde{\xi}) \geq 1 - \lambda_m^* c(\xi, \tilde{\xi}) - \phi_m(\xi), \ \forall \xi, \tilde{\xi}$, we have $\phi_m(\xi) \geq 1 - \tilde{\pi}_m(\tilde{\xi}) - \lambda_m^* c(\xi, \tilde{\xi}), \ \forall \tilde{\xi} \in \widehat{\Xi}$. Hence, we have $\phi_m(\xi) \geq \max_{\tilde{\xi} \in \widehat{\Xi}} \left\{ 1 - \tilde{\pi}_m(\tilde{\xi}) - \lambda_m^* c(\xi, \tilde{\xi}) \right\}$. Recall that $(\pi^*; \lambda_1^*, \ldots, \lambda_M^*)$ is a dual minimizer of problem (10):

$$v_{\text{dual}} := \sum_{m=1}^{M} \lambda_m^* \vartheta_m + \mathbb{E}_{\hat{\xi} \sim \widehat{P}_m} \left[ \max_{\xi \in \widehat{\Xi}} \left\{ 1 - \pi_m^*(\xi) - \lambda_m^* c(\xi, \hat{\xi}) \right\} \right] = \sum_{m=1}^{M} \lambda_m^* \vartheta_m + \mathbb{E}_{\hat{\xi} \sim \widehat{P}_m} \left[ \phi_m(\hat{\xi}) \right],$$

thus

$$v_{\text{dual}} \geq \sum_{m=1}^{M} \lambda_m^* \vartheta_m + \mathbb{E}_{\hat{\xi} \sim \widehat{P}_m} \left[ \max_{\xi \in \widehat{\Xi}} \left\{ 1 - \tilde{\pi}_m(\xi) - \lambda_m^* c(\xi, \hat{\xi}) \right\} \right] = \sum_{m=1}^{M} \lambda_m^* \vartheta_m + \mathbb{E}_{\hat{\xi} \sim \widehat{P}_m} \left[ 1 - \tilde{\pi}_m(\hat{\xi}) \right].$$

Since $v_{\text{dual}}$ is the minimum value, this means that if $\tilde{\pi}$ is a feasible solution, then it is also an optimal solution to (10).

Next we verify $\tilde{\pi}$ is a feasible classifier, i.e., it satisfies $0 \leq \tilde{\pi}(\xi) \leq 1$ and $\sum_{m=1}^{M} \tilde{\pi}_m(\xi) = 1, \forall \xi$. First, by definition, $\pi_m^*(\tilde{\xi}) \geq \tilde{\pi}_m(\tilde{\xi}), \ \forall \tilde{\xi} \in \widehat{\Xi}$, therefore $\tilde{\pi}_m(\tilde{\xi}) \leq 1$ and $\sum_{m=1}^{M} \tilde{\pi}_m(\tilde{\xi}) \leq 1$. If we are able to show that

$$\sum_{m=1}^{M} \tilde{\pi}_m(\tilde{\xi}) \geq 1, \ \forall \tilde{\xi} \in \widehat{\Xi}, \tag{16}$$

then we can show $\tilde{\pi}$ is indeed a feasible classifier.

To show (16), first note that if $\pi_m^*(\tilde{\xi}) = 0$, then we have by definition $\tilde{\pi}_m(\tilde{\xi}) = 0$. Moreover, for any $\tilde{\xi} \in \widehat{\Xi}$, there is a set $\mathcal{M}_0 \subset \{1, \ldots, M\}$ such that for all $m \in \mathcal{M}_0$, $\pi_m^*(\tilde{\xi}) > 0$, and the worst-case

distribution $P_m^*$ transports probability mass from $\mathrm{supp}\,\widehat{P}_m$ to $\tilde{\xi}$, which suggests that there exits $\hat{\xi}_m \in \mathrm{supp}\,\widehat{P}_m$ such that

$$\tilde{\xi} \in \underset{\xi \in \widehat{\Xi}}{\arg\max}\{1 - \pi_m^*(\xi) - \lambda_m^* c(\xi, \hat{\xi}_m)\}.$$

It follows from the definition of $\phi_m$ that

$$\sum_{m \in \mathcal{M}_0} \phi_m(\hat{\xi}_m) = \sum_{m \in \mathcal{M}_0} \left(1 - \pi_m^*(\tilde{\xi}) - \lambda_m^* c(\tilde{\xi}, \hat{\xi}_m)\right).$$

Meanwhile, by definition of $\tilde{\pi}_m$,

$$\sum_{m \in \mathcal{M}_0} \tilde{\pi}_m(\tilde{\xi}) \geq \sum_{m \in \mathcal{M}_0} \left(1 - \lambda_m^* c(\tilde{\xi}, \hat{\xi}_m) - \phi_m(\hat{\xi}_m)\right) = \sum_{m \in \mathcal{M}_0} \pi_m^*(\tilde{\xi}) = 1.$$

Thereby we have shown (16). The proof is completed by noting that the optimal solution $\tilde{\pi}$ satisfies $\|\tilde{\pi}_m\|_{\mathrm{Lip}} \leq \lambda_m^*$ and thus $v_{\mathrm{dual}} \geq v_{\mathrm{Lip}}$. Combine with the previous result that $v_{\mathrm{dual}} \leq v_{\mathrm{Lip}}$, we have shown $v_{\mathrm{dual}} = v_{\mathrm{Lip}}$ and the proof is completed.

$\square$

## C   Memory-efficient implementation of `Dr.k-NN` in data-intensive scenario

For the sake of completeness, we extend our algorithm to non-few-training-sample setting. This can be particularly useful for the general classification problem with an arbitrary size of training set. In fact, $k$-NN methods notoriously suffer from computational inefficiency if the number of labeled samples $n$ is large, since it has to store and search through the entire training set [19].

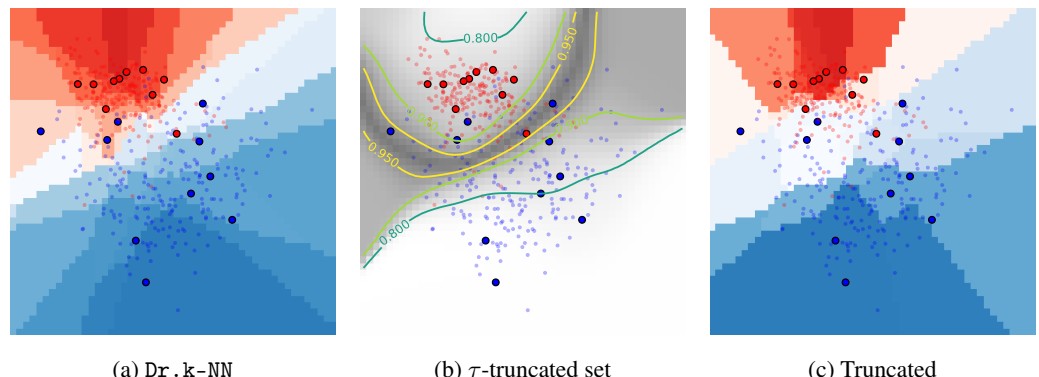

|  (a) `Dr.k-NN` | (b) $\tau$-truncated set | (c) Truncated |

Figure 6: An example of the truncated `Dr.k-NN` using MNIST (digit 4 (red) and 9 (blue)). Big dots represent training samples and small dots represent query samples. (a) shows the decision made by `Dr.k-NN`; (c) shows the decision made by the truncated `Dr.k-NN` with truncation level $\tau = 0.9$; (b) shows $\tau$-truncated regions with $\tau = 0.95, 0.9, 0.8$. Big dots between the lines are selected training samples under different $\tau$. The depth of the shaded area shows the level of samples entropy.

The main idea is to only keep the training samples that are important in deciding the decision boundary based on the maximum entropy principle [11]. As a measure of importance, we choose the samples with the largest entropy across all categories, based on the intuition that the samples with higher entropy has larger uncertainty and will be more useful for classification purposes since they tend to lie on the decision boundary. The entropy of a sample is defined as follows. Consider a random variable which takes value $m$ with probability $\pi_m$, $\sum_{i=1}^M \pi_m = 1$; then the entropy of this random variable is define as

$$H(\pi_1, \ldots, \pi_M) = -\sum_{m=1}^M \pi_m \log \pi_m.$$

As a simple example, for Bernoulli random variable (which can represent, e.g., the outcome for flipping a coin with bias $p$), the entropy function is $H(p) = -p \log p - (1-p) \log(1-p)$, and it

is a concave function achieving the maximum at $p^* = 1/2$, which means that the fair-coin has the maximum entropy; this is intuitive as indeed the outcome of a fair coin toss is the most difficult to predict. Now we use this entropy to define the "uncertainty" associated with each training points. With a little abuse of notation, define

$$H(\widehat{\xi}) := H(\pi_1(\widehat{\xi}), \ldots, \pi_M(\widehat{\xi})).$$

Denote the minimal and maximal entropy of all the training points as

$$H_{\min} = \min\{H(\widehat{\xi}), \widehat{\xi} \in \widehat{\Xi}\}, \quad H_{\max} = \max\{H(\widehat{\xi}), \widehat{\xi} \in \widehat{\Xi}\}.$$

Define the $\tau$-truncated training set as

$$\widetilde{\Xi} = \{\widehat{\xi} \in \widehat{\Xi} : (H(\widehat{\xi}) - H_{\min})/(H_{\max} - H_{\min}) \geq \tau\}, \forall \tau \in [0, 1].$$

The truncated `Dr.k-NN` is obtained similarly as Step 2 of `Dr.k-NN` by restricting the training set $\widehat{\Xi}$ only to the samples in $\widetilde{\Xi}$ (samples with larger entropy). Figure 6 reveals that the most informative samples usually lie in between categories. We can see that a truncated `Dr.k-NN` classifier with $\tau = 0.9$ only uses 20% samples with little performance loss. More experimental details is presented in Section 5.

# D Comparison to kernel smoothing

Figure 7 and Figure 8 present a comparison of the results using `Dr.k-NN` and the kernel smoothing defined in (13). The results suggest that the performance of `Dr.k-NN` is insensitive to the choice of $k$, while the performance of the kernel smoothing is heavily depended on the choice of $h$.

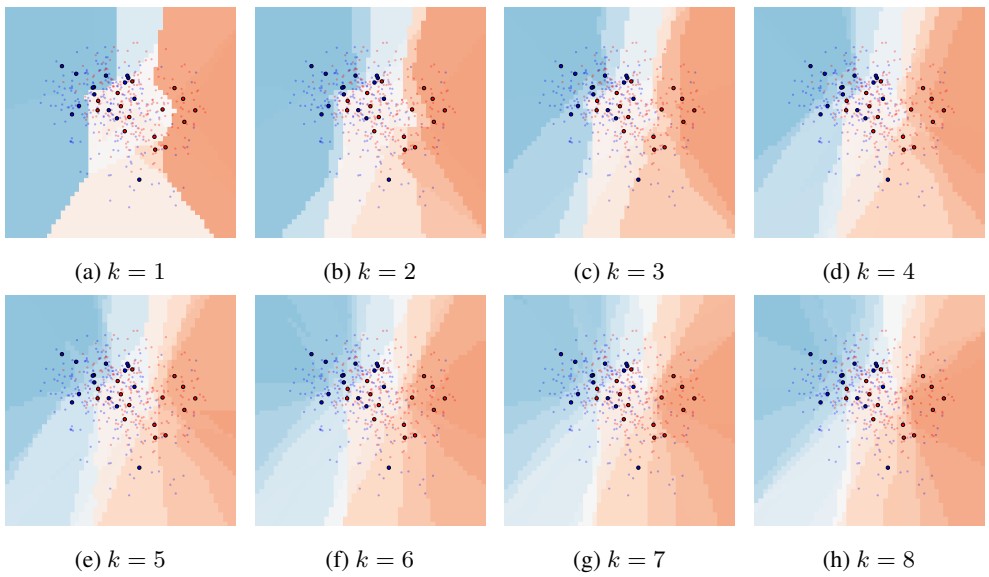

(a) $k = 1$     (b) $k = 2$     (c) $k = 3$     (d) $k = 4$

(e) $k = 5$     (f) $k = 6$     (g) $k = 7$     (h) $k = 8$

Figure 7: `Dr.k-NN` with different $k$.

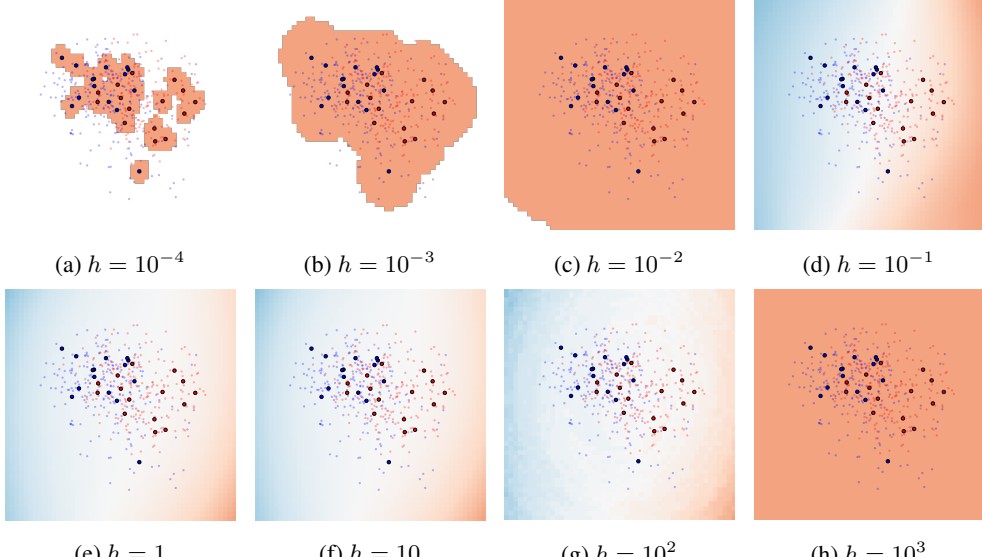

(a) $h = 10^{-4}$     (b) $h = 10^{-3}$     (c) $h = 10^{-2}$     (d) $h = 10^{-1}$

(e) $h = 1$     (f) $h = 10$     (g) $h = 10^2$     (h) $h = 10^3$

Figure 8: Kernel smoothing with different bandwidth $h$.

# E    Real data examples for COVID-19 CT

Figure 9 and Figure 10 show 16 real CT images collected from patients who have been diagnosed with COVID-19 and other diseases (non-COVID-19), respectively.

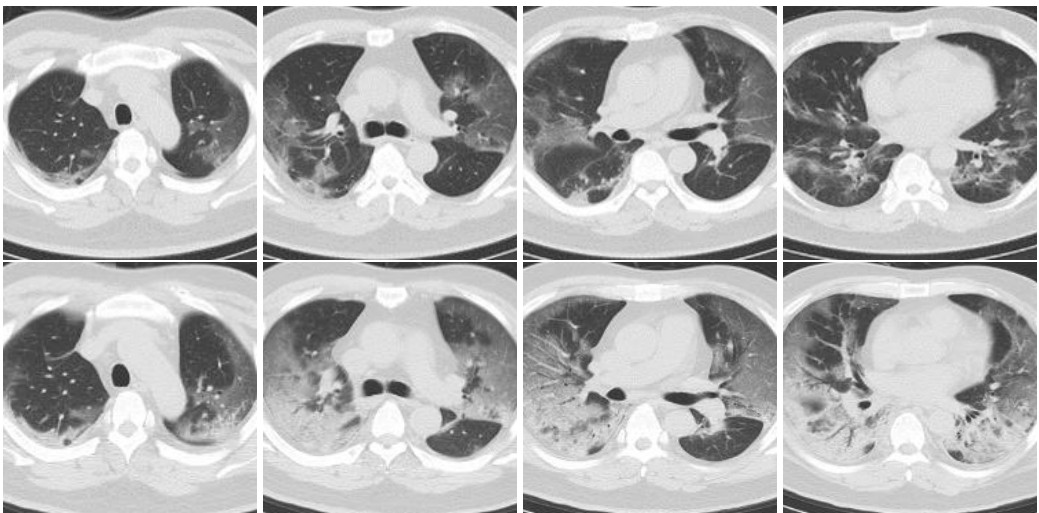

Figure 9: COVID-19 CT images.

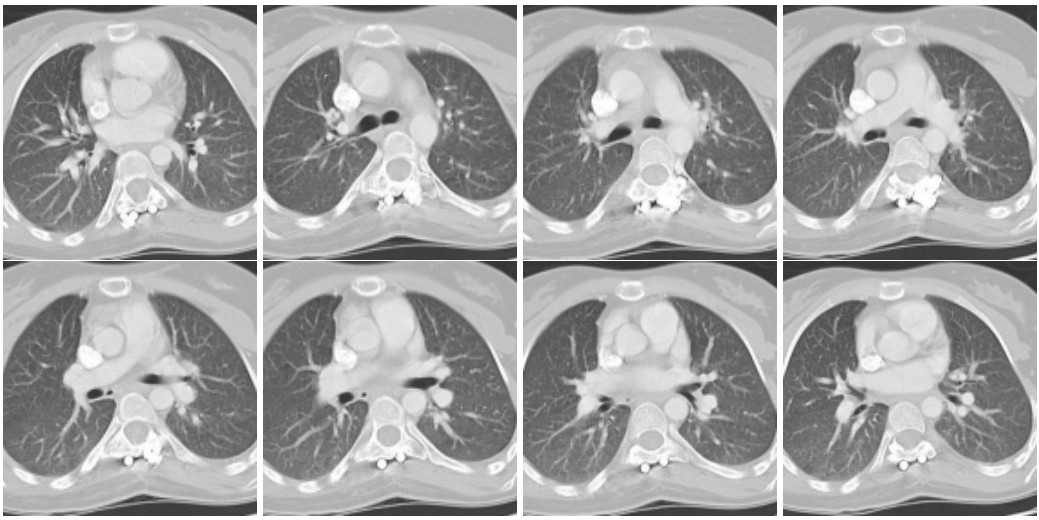

Figure 10: Non-COVID-19 CT images.