# OpenReview forum: "Distributionally robust weighted k-nearest neighbors"
_NeurIPS.cc/2022/Conference — NeurIPS 2022 Accept_

### Official Review · Reviewer_Spo7 · 2022-07-11

**Rating:** 5
**Confidence:** 4
**Soundness:** 2 fair
**Presentation:** 3 good
**Contribution:** 2 fair

**Summary:**

This paper aims to developing a distributionally robust KNN classifier for multiclass few shot scenario to mitigate those weaknesses of existing similar methods, it essentially learns the class-dependent metrics to build corresponding optimal weighted k-NN classifiers, , so-designed algorithm Dr. KNN is able to hedge against feature uncertainties.so-reported comparison results show relatively favorable performance to SOTAs..

**Questions:**

1.Existing works should NOT be omitted in the following aspects
In few-shot metric learning:
[m1]Wen Jiang, et al, Multi-Scale Metric Learning for Few-Shot Learning, IEEE Transactions on Circuits and Systems for Video Technology, 2021,Page(s):1091 - 1102.
[m2] Zheren Fu, etal, "Self-Supervised Synthesis Ranking for Deep Metric Learning", IEEE Transactions on Circuits and Systems for Video Technology, vol.32, no.7, pp.4736-4750, 2022.
[m3]X Li, etal, Deep metric learning for few-shot image classification: A selective review,  arXiv preprint arXiv:2105.08149, 2021.
In robust metric learning:
[m4]K Huang, et al, Robust metric learning by smooth optimization, arXiv preprint arXiv:1203.3461, 2012.
[m5] L Wang, et al, Provably robust metric learning, NerISP2020.
[m6]Blanchet, et al, Doubly robust data-driven distributionally robust optimization,J  arXiv preprint arXiv:1705.07168, 2017.

2' "the importance of choosing a good weighting scheme in weighted k-NN" is natural, however, In an example in Figure 1, whether do the colors and shapes of samples also play some crucial roles in classifying them?
Especially for a few shot scenario, if we do NOT take these, including disentanglement for the building blocks of samples, into account but just learning some metric, how far can we go? implying just doing these seems NOT far enough, for example, one can simply change the color of test samples with the same training class to outside those of theirs, is it still robust?
thus the distributions you assume will significantly introduce bias to learning, which, undoubtedly, also brings robustness in other sense!

3. Can the method or designed algorithm handle imbalanced or/and large classes? In your experiments, you just adopt the maximal class number M=5, suggesting to increase it for more comparison!



**Limitations:**

Though the formulation or definition in this manu. is somewhat trivial, but its highlight lies in optimization and theoretical property analysis from which some conclusions or insights can be gained.

**Strengths And Weaknesses:**

Strengths:
1.For multi-class few shot metric learning, the authors develop the optimal weighted k-NN classifiers by using the proposed Dr. KNN algorithm to optimize a defined distributionally robust formulation, including the class-dependent weights in classification.
2.Theoretically proving the formulation equivalent to a Lipschitz norm regularization problem and analyzing a few properties to justify their algorithm;
3.Empirically, confirming the proposed algorithm to have competitive performance compared to the SOTAs in the same setting with various real-data sets.

Weaknesses
1.The proposed formulation lacks a sufficient clarification about the uniqueness, including, essential difference in principle from existing DRO formulation.
2.the problem under study involves distributional robustness and metric learning, thus the authors should NOT overlook some existing works in the two aspects, at least being mentioned to make differences, especially those appeared in 2021 and 2022.
3.The assumption among classes is NOT practice.

---

> ### Author Response · Authors · 2022-08-02
> **Response to Reviewer Spo7**
>
> We thank the reviewer's positive comments and provide our response below:
> - We thank the reviewer's comment and add the following description: The proposed formulation is a unique and tailored setup for the robust k-NN problem. Compared with existing DRO formulation, we highlight some key differences which requires new analysis techniques: (i) we have $M$ ambiguity sets in our formulation caused by the multi-class nature of this problem; (ii) our objective function is related to the $M$ mis-classification errors, meaning that those $M$ ambiguity sets cannot be viewed independently, thus making the LFDs challenging to solve; while in standard Wasserstein DRO problems, there is only one ambiguity sets and only one worst-case distribution to be found. Moreover, the idea of using the LFDs to construct a weight function for weighted $k$-NN classifier is also novel to our best knowledge.
> - We thank the reviewers for pointing out these literature. We will add the related literature and discussion into the revised version. For example, we found the some of the most recent literature are indeed closely connected to our study, including: Liao, Tingting, et al. "Deep Metric Learning for K Nearest Neighbor Classication." IEEE Transactions on Knowledge and Data Engineering (2021); Pal, Anabik, et al. "Deep metric learning for cervical image classification." IEEE Access 9 (2021): 53266-53275; Chen, Ruidi, and Ioannis Ch Paschalidis. "Distributionally robust learning." Foundations and Trends® in Optimization 4.1-2 (2020): 1-243.
> - If we understand your question correctly, we recognize that our method is required to have at least one training sample from each class to generate a reasonable minimax formulation, which is different from metric learning. But other than this, we believe we have rather weak assumptions for the underlying true distributions of each class, and particularly, we do not impose additional conditions on the data distributions for each class. We are willing to provide more details if you can elaborate more on your concerns.
>
> We also thank the reviewer's suggestions and answer the questions below:
> - We thank the reviewers for pointing out these literature, and we will add them in the revised version.
> - We want to clarify that every details of the image in the example of Figure 1, including color and shape, plays an important part in classifying them. In our Dr. kNN framework, these information will be captured by the feature embedding represented by the deep neural network (e.g., CNN in our experimental setting); the "distance metric" or the weights we learned is in fact in the feature space, telling us how important these features are under different categories. See Figure 3 for the overall pipeline -- we will first use neural networks (e.g., CNN) to produce efficient feature embedding for images and then apply the robust classification framework. We also want to emphasize that one of the most important contributions of this work is this framework can be trained in an end-to-end fashion (the gradients can be back-propagated through the optimization layer to the weights of the neural network).
> - We thank the reviewer for the constructive suggestions. Here we first briefly state the computational complexity of the dr.knn algorithm. Compared to vanilla k-NN, our proposed method involves solving an additional convex optimization eq (9), which has $n^2M$ variables and $(2n+1)M$ constraints. It is a linear program when $c$ is 1-norm or $\infty$-norm, and a conic program when $c$ is 2-norm. In the few-sample regime, it is computationally efficient by adopting the differentiable convex optimization layers. And the complexity {\it linearly} depends on $M$. We also present a truncated version of Dr. kNN in the supplementary material, which enables the algorithm to process a larger scale of training data. This algorithm can be also extended to the large classes scenario. Lastly, we will add more ablation studies and comparisons with larger number of classes in our future work.
>
> Lastly, we want to thank the reviewer for the positive comments in the limitations. We would like to emphasize that our formulation and analysis adds non-trivial contribution to the DRO literature as we pointed out in the response to the weakness of the paper.

---

> > ### Comment · Reviewer_Spo7 · 2022-08-08
> > **Thanks the authors for their responses to most of my concerns**
> >
> > Indeed, the authors have answered most questions of mine, however, there are still 2 concerns to be solved as follows:
> > 1. For missing existing works, I currently fail to decide the advantages of your learning algorithms after all no enough evidence is given.
> > 2. For so-learnt metrics, due to few-shot scenario, the metrics more likely do not learn some features NOT appearing in the test objects as has been mentioned previously, for example, the same object has different colors in which some appear in training but some JUST appear in test data, which is real and natural, implying your metrics will fail and thus involving some implicit assumption! For this, I hope I have explained clearly enough!

---

> > > ### Author Response · Authors · 2022-08-08
> > > **Thanks the reviewer for the feedback**
> > >
> > > Dear Reviewer,
> > >
> > > We thank you for your feedback and please see our responses in the following:
> > >
> > > (1) As we mentioned in the introduction, we want to clarify that we focus on attacking the general multiclass-classification problem with few training samples, which indeed shares some similarities with few-shot learning. However, the main difference between these two is that we assume there is no unobserved class. In other words, we do not acquire meta knowledge from a large number of observed classes and then predict examples from unobserved classes. Therefore, most of the few-shot learning methods can not be directly compared with our proposed approach, and we only provides comparisons with three major seminal few-shot learning works (prototypical, matching net, and metaoptnet) for the sake of completeness. To facilitate understanding of our major contributions, we also summarize the advantage of our learning algorithms in the following three perspectives: (i) The proposed dr.knn algorithm significantly outperforms other baselines with respect to the average test accuracy on all data sets, as we have shown in Table 1. We also compare the decision boundary of dr.knn with other methods in Figure 5, to validate that the proposed learning framework will lead to better feature representation with a smooth decision boundary and a reasonable decision confidence map. (ii) The proposed learning algorithm is computationally efficient and easy to carry out as shown in Algorithm 1 (Appendix A). Also, the final performance of our method is insensitive to the selection of the only hyper-parameter $k$ in our framework. (iii) Theory-wise, this is the first work to study the distributionally robust $k$-nn classifier with Wasserstein ambiguity sets. Tractable reformulations and theoretical guarantees are both provided.
> > >
> > > (2) Because, we focus on attacking the general multiclass-classification problem with few training samples instead of few-shot learning, we agree with the reviewer and respect the point that it would be universally challenging to tackle few-training-sample problems if some of the features in the test data do not appear in the training data no matter what types of methods are adopted. However, the experimental results tell another story and show that our method did not fail completely even when some of the feature are missing. For example, in COVID-19 and Lung Cancer data sets, the pattern of being diagnosed as having cancer or covid-19 is very intricate and can certainly not well covered by the training data. Specifically, we only observe very little high-dimensional samples (5 to 10 data samples per class), whereas the size of the testing data pool is relatively large (more than 300). It turns out that our method significantly outperforms other baselines on both of two real data sets. One possible reason to explain the performance is that our deep feature extractor $\phi(;\theta)$ can capture the key structure of the feature from the limited training samples, where the samples with minor differences (for example, in different color) are closer in the feature space; Another reason is that, because of the nature of our DR method, the algorithm is likely to assign lower confidence (or weight) to the category for the query sample if they are dissimilar in the key feature that  distinguishes the category from others.
> > >
> > > Again, we would like to appreciate the reviewer's constructive feedback, and will add more ablation studies in the future to further improve the quality of our paper.

---

### Official Review · Reviewer_PP67 · 2022-07-11

**Rating:** 7
**Confidence:** 4
**Soundness:** 3 good
**Presentation:** 4 excellent
**Contribution:** 3 good

**Summary:**

The authors take the generalization of the K-NN method for the multi-label classification problem, which lifts the samples to feature spaces and replaces the distance weights with more general weight functions. The distributionally robust formulation of this well-known generalization is defined and shown to be equivalent to a much simpler problem when the ambiguity sets comprise Wasserstein balls. Thanks to this equivalence, the authors show that the worst-case distributions are characterized by the solution of a convex optimization problem. There is further a solution algorithm proposed, and thanks to this, the authors compare the performance of Wasserstein DRO weighted K-NN with benchmark algorithms on well-known classification datasets.

**Questions:**

I would like to ask the following questions:

- The authors find that their approach is identical to Lipschitz regularization. To what extent is this finding in line with your citation [30]? It is already known that Wasserstein DRO is identical to regularization in general classification settings. Specifically: (i) if the authors believe this finding is novel and not directly applicable from [30] (e.g., because here we optimize a probability vector) could they please discuss this (ii) if it indeed follows from [30], it is still interesting, because [30] shows equivalence with *a* regularizer, but here the authors show a specific one.

- Page 3: randomized "test" -> what does 'test' mean?

- Page 3, line 128: Suppose the features in each class $m$ follows a distribution $P_m$ -> could the authors please state this more formally?

- Page 4, lines 160-161, the optimization problem is written in a way that it looks like $m$ is also an optimization variable.

- Could the authors please explain the derivation of (8) intuitively -> i.e., by restricting problem (6) via extra constraints (which is then shown to be redundant in terms of the optimal solution)

- Page 6, line 212 states usage of "duality for Wasserstein DRO" -> this does not always hold and there needs to be some structure for duality. Could the authors please mention them?

**Limitations:**

Overall, I am positive about the paper. I would like to clarify the questions I asked above, as well as the weaknesses mentioned.

My biggest concern (or question that I would like to clarify) is that the authors constrain the ambiguity sets to include distributions that are supported only on the training instances. In general, in most Wasserstein classification settings, the most useful results are thanks to the fact that we do *not* have such constraints. It can be seen from the literature that the worst-case distributions are (typically supported on at most $n+1$ atoms -- please also check if this holds here) characterized by a weighted mixture of training points, as well as a point that is extremely far away from the training points, though with a negligibly small weight. This is how the Wasserstein methods coincide with regularization techniques. Would it be possible for the authors to compare their method with a brute-force method that solves the Wasserstein DRO problem where the ball's support points are unconstrained?

I am also wondering whether the equivalence between (6) and (8) works because of such an assumption. If my concern is not valid, I would appreciate an explanation from the authors.

**Strengths And Weaknesses:**

Strengths:
The paper is written extremely well. It is very easy (and fun) to follow. The motivation is clear. The proofs are correct and they follow a modern set of techniques. The numerical experiments are very thorough and interesting.

Weaknesses:
- There are some missing discussions about the Wasserstein DRO side of the paper. Especially, recently, there is a strong focus on the structure of the worst-case distributions, finite sample guarantees, and asymptotic consistencies. These are not mentioned in this paper, and except for defining and solving the problem, there is not much focus on the properties that come thanks to the Wasserstein formulation.
- The ambiguity sets (that said, the authors call those uncertainty sets, which I believe should be named ambiguity sets) are restricted to distributions supported on training points. I have never seen this, and this may be a dangerous approach. I would like to see more discussions on this. If I am wrong, then seeing further references would be great.

Further details are in the "Limitations" section.

---

> ### Author Response · Authors · 2022-08-02
> **Response to Reviewer PP67**
>
> We appreciate reviewer's positive comments and see our response below
> - We thank the reviewer for pointing out the missing discussion for Wasserstein DRO. Our solution and analysis to the robust k-NN problem indeed relies on some important computational and theoretical properties of Wasserstein DRO, like the least favorable distributions and the connection with Lipschitz regularization problems. Although the asymptotic consistencies may not hold here because we focus on the small-sample-size regime where robustness is crucial, but the finite-sample guarantees are helpful and can be helpful for radius choice. In addition, the worst-case distribution, what we call least favorable distributions in this work, also enjoys similar structure as in Wasserstein DRO, reflected in e.g., the dual form in equation (10).
> - Our main motivation in formulating the problem is to use the least favorable distributions (LFDs) resulting from the Wasserstein DRO problem to construct the weight function for weighted $k$-NN. Because a $k$-NN algorithm makes its decision solely based on the training samples, to ensure the LFDs supported on the training samples only, we devise the ambiguity set with distributions supported on the training sample, so that the corresponding LDFs can be used as the corresponding weights under different categories. If we do not restrict the support of the considered distributions, for binary classification problems, one can still prove that the LFDs are supported on the training samples, see the reference [15] of the submission. On the other hand, for more than two classes, if we do not restrict the support, then there are example where the LFDs put positive probability other than the training sample, in which case one cannot construct a meaningful weighted $k$-NN based on the weights of LFDs. We also remark that other DRO literature sometimes also restrict the support, for example, when considering stress test problems [Blanchet and Kang 2021].
>
> We also thank the reviewer's suggestions and answer the questions below
> - Compared with existing results, e.g., [13] and [30] on connection between DRO and regularization, our finding is novel in several aspects: (i) Classical results typically focus on one Wasserstein ambiguity sets, while in our cases we have multiple ambiguity sets, (ii) Classical results rely on the assumption that the objective function is continuous on a continuous space, while the mis-classification errors in our objective function is 0-1 loss on training samples; (iii) Indeed, we give a specific regularizer in our setting.
> - We apologize for the confusion. We meant randomized ``classifier'' and deterministic ``classifier'' in page 3. The main difference is that the classification result of the random ``classifier'' is random according to certain distributions.
> - Formally, we suppose the feature (data) $x$ in each class/label $m$ follows a distribution $P_m:=P(x|y=m)$;
> - We apologize for the confusion. $m$ is not an optimization variable, we meant that $P_1,\ldots,P_M$ are the optimization variables. We will revise the optimization problem into $\max_{P_1\in\mathcal P_1,\ldots,P_M\in\mathcal P_M} \Psi(\pi; P_1, \dots, P_M)$;
> - The minimax problem defined in (8) is a very general minimax classification problem, where the search space for the classifier is the entire space of all random classifiers. Therefore, the robust k-NN problem is just a subproblem of eq(8) because the search space in eq(6) is restricted to robust k-NN classifiers. The reason we introduce (8) is that it enjoys a nice finite-dimensional convex reformulation problem as shown in eq(9), thus much easier to solve. Moreover, we show in Theorem 2 that although the robust k-NN has a smaller search space, it can indeed achieve the same optimum value as eq(8) (and thus the solution to eq(9) is also the optimal solutioin to eq(6)).
> - Yes, we mainly use Theorem 1 [14] and require the following conditions for the Wasserstein DRO to hold: (i) the Wasserstein distance is of order $p\geq 1$ (satisfied as we use $p=1$); (ii) the objective function $\Psi$ defined in eq(1) to be an $L^1$ function on the nominal measure $\hat P_1,\cdot, \hat P_M)$ (the center of the ambiguity sets), this is satisfied obvious as we have $\Psi\leq M$ is a bounded function, and it is $L^1$ on any probability measures.
>
> We also thank the reviewer for raising the point in limitations. We restrict the support of the distributions in the ambiguity set so that the weighted $k$-NN scheme resulting from the LFDs are well-defined. Therefore, we were not able to compare with an alternative formulation that do not restrict the support of the distribution even with brute-force solution method, and $\pi^{knn}$ in (6) will no longer be well-defined without such an assumption. Moreover, given the connection with a new regularizer, we believe the finding is still interesting when restricting the support.
>
> We will revise the text accordingly based on our response.

---

> > ### Comment · Reviewer_PP67 · 2022-08-08
> > **Acknowledging the rebuttal**
> >
> > Thank you for your thorough reply. I appreciate this summary and apologies that my initial review had a strong focus on Wasserstein ambiguity sets. I agree that the work is more general.
> >
> > I also read other reviewers' responses and the authors' replies. Overall, I am staying positive about this paper.
> >
> > I have one final question:
> > - Is it true that in the numerical experiments currently none of the benchmark methods are distributionally robust (DR) classification methods? In that case, the only DR method is the one proposed, which might perhaps not be fair to existing work. In my view, the fact that this paper is specifically working with the few-sample case does not prevent trying general DR models, and it would be great to see how they compare. This is a minor point as the paper is in the KNN universe anyway but just wanted to ask.

---

> > > ### Author Response · Authors · 2022-08-08
> > > **Thanks the reviewer for the feedback**
> > >
> > > Dear Reviewer,
> > >
> > > Thank you so much for your suggestions and it is true that we mainly compare the proposed dr.knn algorithm with existing knn-based classifiers. We are indeed aware of some recent advancements in developing DR classifiers, such as distributionally robust logistic regression, and agree with the reviewer that it would also very interesting to compare our method with these exiting DR classifiers. We are planning to add these comparisons in our future work as another extended study.
> > >
> > > In the meantime, we also want to clarify the two major reasons why we didn't compare with DR classifiers in this study in the first place: (1) As you commented, we mainly focus in the $k$-nn scheme, and our robust classifier can be viewed as a weighted $k$-nn, with weighted selected in a DR way. Therefore, we mainly compare the performance with other (weighted) knn classifiers to show the advantage of dr.knn in the small-sample-size regime; (2) In this study, we are targeted at a general multi-classification problem under non-parametric setting, where we do not make strong assumptions for the data structure / distribution. As a result, $k$-NN algorithms are a natural fit and have been widely adopted for addressing this type of problems. In contrast, most of DR classifiers requires parametric model assumptions, which applies to specific tasks varying from application to application. For different applications, the choice of their models and the selection of their hyper-parameters are another challenging task, which would very likely affect their final performances significantly and go beyond the scope of this paper. Therefore, we didn't include these methods into our baselines in the experiments.

---

> > > > ### Comment · Reviewer_PP67 · 2022-08-08
> > > > **Thank you for the reply**
> > > >
> > > > Dear Authors,
> > > >
> > > > Thank you for your reply. I understand your motivation. Thanks for the explanation.
> > > >
> > > > Best regards,
> > > > Reviewer PP67

---

### Official Review · Reviewer_PjNq · 2022-07-12

**Rating:** 8
**Confidence:** 3
**Soundness:** 4 excellent
**Presentation:** 4 excellent
**Contribution:** 4 excellent

**Summary:**

This paper proposes a distributionally robust version of k-nearest neighbors (k-NN) classifier that can perform well in a small-sample regime, especially for a multiclass setting.
The authors propose to consider a minimax optimization problem for a distributionally robust classification, and show that this infinite-dimensional problem can be indeed solved by a finite-dimensional convex problem.
Its connection to the Lipschitz regularization framework is also established.
They then propose the Dr. k-NN algorithm and show that it can be seamlessly used with learning neural features jointly.
The experiments show that the proposed algorithm can beat the existing baselines as well as other neural network based approaches.

**Questions:**

- In line 139, after "In the sequel, we deﬁne a general tie-breaking rule as follows." I do not find a tie-breaking rule.
- In Table 1, in the column for Omniglot, $M=2$, $K=5$, Prototypical Net achieves 0.769, which is the highest, but it's not highlighted.
- In line 101, it is claimed that "we analyze the generalization bound." Is the connection to the Lipschitz regularization problem the referred "analysis" of the generalization bound? If that is the case, I would suggest to clarify the sentence to avoid any misunderstanding.
- A closely related work of a similar flavor missed in the current paper is:
    - Farzan Farnia and David Tse, "A Minimax Approach to Supervised Learning," NeurIPS 2016.

   This paper also studied the same minimax optimization framework with a similar motivation, but under a different form of the candidate set of distributions. It would be helpful to discuss the similarities and differences.

**Limitations:**

It is indicated in Checklist that limitations are mentioned in Section 6, but I cannot find any.
Is there any limitation of this framework?

**Strengths And Weaknesses:**

This is a solid and well-written paper.
The mathematical formulation and the technical results are well motivated and very elegant based on the convex optimization theory.
The presentation is also very clear except that the notation is a bit heavy.
The experiments are well designed and executed to corroborate the power of the theoretical framework.

Weaknesses are hard to find.

---

> ### Author Response · Authors · 2022-08-02
> **Response to Reviewer PjNq**
>
> We very much appreciate reviewer's positive comments and now provide our response on a point-by-point basis:
>
> - 1. We apologize for the confusion caused. To deal with cases with ties, we aim to use a very general tie-breaking rule and we do not impose additional constraints at this stage. We describe necessary notations for tie-breaking rule in Line 141-143, which means that we break the tie arbitrarily among the classes $\mathcal M_0(\xi)$ on which the corresponding weights $p_m(\xi)$ is maximum.
> - 2. We thank the reviewer for pointing this out. We now have fixed this in the revised version.
> - 3. Yes the connection to the Lipschitz regularization problem is the main technique for the generalization analysis in this paper; and once we make the connection with the Lipshitz regularization problem, we apply existing generalization analysis for Lipchitz functions. We thank the reviewer's suggestions and we have revised the sentence in Line 101 to ``Third, we analyze the generalization bound by proving the connection with Lipshitz regularization problem''.
> - 4. We thank the reviewer's suggestions and have added the paper to the references; it indeed has a similar motivation as the minimax optimization framework. We will add more discussion on the related literature in the revised version. To be specific, [Farnia and Tse 2016] aims to address the general supervised learning by introducing a generalization of the maximum entropy principle. The proposed approach solves a minimax optimization problem and is shown to be equivalent to the regularization problem, which are both similar to ours. However, our work differs from this paper from the following aspects: First, rather than attacking a general supervised learning task, we focus on a particular type of classification problem, where only a few training samples are available. Second, since the decision making relies on a weighted $k$-NN algorithm, our objective is to find a weight vector (LFD) for each sample that optimizes the performances under the worst scenario. Third, as mentioned by the reviewer, [Farnia and Tse 2016] considers a different type of ambiguity sets defined via cross-moments constraints, while our sets are Wasserstein sets, this leads to different tractable optimization reformulation and different regularization terms.  Last but not least, we introduce an end-to-end learning framework that enhances the power of our method by taking advantage of deep neural network as the feature extractor.

---

### Official Review · Reviewer_67DR · 2022-07-14

**Rating:** 5
**Confidence:** 3
**Soundness:** 3 good
**Presentation:** 4 excellent
**Contribution:** 2 fair

**Summary:**

This paper studies the k-NN algorithm when applied to multiclass classification with a few samples. The authors develop algorithms by formulating a  distributionally robust variant of k-NN where each nearest neighbor is weighted based on least favorable distribution.



**Questions:**

NA

**Ethics Review Area:**

["I don’t know"]

**Limitations:**

The paper lacks novelty and empirical evaluation does not strongly support the superior performance of the proposed method.


**Strengths And Weaknesses:**

The paper is well-written, and the proposed algorithm is supported by theoretical results and empirical evaluation. However, it lacks novelty and empirical evaluation does not convey the superior performance of the proposed method. For example, the authors discuss the success of metric learning in kNN but exclude it in the experiments. I understand that in small per-class setting, the similar and dissimilar sets for learning the distance metric would be very imbalance, but it would interesting to see to what extent the proposed algorithm improves upon it (by making tweaks to metric learning such as hard negative sampling etc to overcome the imbalance ness issue). Overall, while the paper is very well-written and enjoyable to read, the lack of novelty and aforementioned issue on empirical evaluations prevents me from giving it a high score.

---

> ### Author Response · Authors · 2022-08-02
> **Response to Reviewer 67DR**
>
> We greatly thank the reviewer for the positive feedback and very much appreciate the constructive suggestion. First, we would like to mention that the main methodological contribution of this paper is to propose a new weighted robust $k$-NN algorithm. We emphasize several key novelties below:
>
> - First, in the propose framework, each sample is associated with a tailored weight vector rather than a scalar value in classical k-NNs. This is a brand new weighting scheme in the sense that we do not only care about the individual importance of each sample, but also how its importance (and most importantly, ``uncertainty'') is reflected under different categories/classes.
> - Second, although our main formulation in Eq (6) and Eq (8) look similar with existing Distributionally Robust Optimization (DRO) settings, they are fundamentally different because here multiple ambiguity sets are involved.  And the mis-classification errors, one for each ambiguity set, are dependent on each other, making the theoretical analysis challenging and new techniques need to be developed.
> - Theory-wise, our proof for Theorem 1 and 2 is one of the most important theoretical results in this paper, showing that the original infinite-dimensional functional optimization over all weighted $k$-NN classifiers can be solved efficiently via a finite-dimensional convex optimization program.
> - Moreover, we also make the connection with Lipschitz norm regularization problem (and the regularizer are given specifically), which enables us to study the generalization performance of the robust classifier.
>
> Secondly, we thank the reviewer for the great question and want to clarify that, instead of directly learning the "distance metric", the goal of our framework is to learn a least favorable distribution (LFD) for each sample over different categories by solving a minimax optimization problem, which does not need to compare similar and dissimilar sample pairs in the training data, and therefore, circumvents the imbalance issue. Intuitively, more "similar" samples (cluster in the feature space) with the same category usually means less uncertainty if the query is close to them. On the flip side, we recognize that our method is required to have at least one training sample for each category to generate reasonable results because of our minimax formulation.
>
> Lastly, we would also like to clarify that we did compare the proposed method with other mainstream metric learning approaches, including NCA, matching net, prototypical net. The results show that our method significantly outperforms other baselines under the few-training-samples setting.

---

### Author Response · Authors · 2022-08-08
**Request for further feedbacks**

Dear Reviewers,

We would like to thank you all for taking the time and effort to review the manuscript and constructive comments, which helped us to improve the quality of the manuscript. We have carefully provided our responses in the comments and look forward to your further feedbacks.

Best,
Authors

---

### Meta-Review · Area_Chair_iFvT · 2022-08-30

**Recommendation:** Accept
**Confidence:** Less certain

**Metareview:**

The reviewers conclude on an interesting paper (especially PjNq) with substantial results that justify its acceptance. I can only recommend to include all of the discussion parts in the camera ready version.

**Award:**

No

---

### Decision · Program_Chairs · 2022-09-14

Accept